**Highly time-resolved characterization of carbonaceous aerosols using a two-wavelength**
**Sunset thermal-optical carbon analyzer**
Mengying Bao[1,2,3], Yan-Lin Zhang[1,2,3*], Fang Cao[1,2,3], Yu-Chi Lin[1,2,3], Yuhang Wang[4], Xiaoyan
Liu[1,2,3], Wenqi Zhang[1,2,3], Meiyi Fan[1,2,3], Feng Xie[1,2,3], Robert Cary[5], Joshua Dixon[5] and Lihua
Zhou[6]
*1 Yale-NUIST Center on Atmospheric Environment, Joint International Research Laboratory*
*of Climate and Environment Change (ILCEC), Nanjing University of Information Science and*
*Technology, Nanjing 210044, China*
*2 Key Laboratory of Meteorological Disaster Ministry of Education (KLME), Collaborative*
*Innovation Center on Forecast and Evaluation of Meteorological Disasters (CIC-FEMD), Nanjing*
*University of Information Science and Technology, Nanjing 210044, China*
*3 School of Applied Meteorology, Nanjing University of Information Science and Technology,*
*Nanjing 210044, China*
*4 School of Earth and Atmospheric Sciences, Georgia Institute of Technology, Atlanta 30332,*
*USA*
*5 Sunset Laboratory, 1080 SW Nimbus Avenue, Suite J/5 Tigard, OR 97223, USA*
*6 College of Global Change and Earth System Science, Beijing Normal University, Beijing*
*100875, China*
*Correspondence: Yan-Lin Zhang (dryanlinzhang@outlook.com)*
**Abstract**
Carbonaceous aerosols have great influence on the air quality, human health and climate
change. Except for organic carbon (OC) and elemental carbon (EC), brown carbon (BrC) mainly
originates from biomass burning, as a group of OC with strong absorption from the visible to near-
ultraviolet wavelengths and makes a considerable contribution to global warming. Large amounts
of studies have reported long-term observation of OC and EC concentrations throughout the word,
but studies of BrC based on long-term observations are rather limited. In this study, we established
a two-wavelength method (658 nm and 405 nm) applied in the Sunset thermal-optical carbon
analyzer. Based on a one-year observation, we firstly investigated the characteristics,
meteorological impact and transport process of OC and EC. Since BrC absorbs light at 405 nm
more effectively than 658 nm, we defined the enhanced concentrations (dEC $=EC_{405\,nm}—EC_{658\,nm}$)
and gave the possibility to provide an indicator of BrC. The receptor model and MODIS fire
information were used to identify the presence of BrC aerosols. Our results showed that the
carbonaceous aerosol concentrations were highest in winter and lowest in summer. Traffic
emission was an important source of carbonaceous aerosols in Nanjing. Receptor model results
showed that strong local emissions were found for OC and EC; however, dEC was significantly
affected by regional or long-range transport. The dEC/OC and OC/EC ratios showed similar
diurnal patterns and the dEC/OC increased when the OC/EC ratios increased, indicating strong
secondary sources or biomass burning contributions to dEC. Two biomass burning events both in
summer and winter were analyzed and the results showed that the dEC concentrations were
obviously higher in biomass burning days; however, no similar levels of the OC and EC
concentrations were found both in biomass burning days and normal days in summer, suggesting
that biomass burning emission made a great contribution to dEC and the sources of OC and EC
were more complicated. Large number of open fire counts from the northwest and southwest areas
of the study site were observed in winter and significantly contributed to OC, EC and dEC. In
addition, the near-by Yangtze River Delta (YRD) area was one of the main potential source areas
of dEC, suggesting that anthropogenic emissions could also be important sources of dEC. The
results proved that dEC can be an indicator of BrC in biomass burning days. Our modified two-
wavelength instrument provided more information than the traditional single-wavelength thermal-
optical carbon analyzer and gave a new idea about the measurement of BrC; the application of
dEC data needs to be further investigated.
**1. Introduction**
Carbonaceous aerosols including organic carbon (OC) and elemental carbon (EC), which
have significant influence on the global radiative transfer, human health and atmospheric visibility,
have been the focus of research in the atmospheric environment field for many years (Lelieveld et
al., 2015; Wu and Yu, 2016; Wang et al., 2018; Zhang et al., 2017; Liu et al., 2019; Zhang et al.,
2019). EC mainly originates from fossil fuel and biomass combustion and is estimated to be the
second largest warming factor behind $CO_2$ contributing to climate change (Liu et al., 2015; Zhang
and Kang, 2019; Cao and Zhang, 2015). OC originates both from primary emissions and gas-to-
particle conversion as secondary organic carbon (SOC) and can scatter the solar radiation which
causes negative forcing globally (Zhou et al., 2014; Huang et al., 2014).

In the recent decades, brown carbon (BrC), as a kind of light-absorbing organic carbon which can absorb light especially from near-UV to visible wavelength, has caused global concern due to its positive climate effect (Andreae and Gelencsér, 2006; Zhang et al., 2020). BrC is mainly emitted from anthropogenic and biogenic emissions (Zhang et al., 2011). Previous studies have proved that biomass burning and biofuel combustion are the most important sources of primary BrC (Saleh et al., 2014; Wu et al., 2020; Lei et al., 2018). Recent researches reported that in developing countries such as China and India, the contribution of fossil fuel combustion to BrC cannot be ignored (Satish et al., 2017; Yan et al., 2017; Kirillova et al., 2014). Secondary BrC is mainly produced by heterogeneous photo-oxidation reactions or aqueous reactions of anthropogenic and biogenic precursors (Zhang et al., 2020; Li et al., 2020; Zhang et al., 2011). However, due to the lack of understanding of BrC at the molecular level and in situ BrC data, there are still large uncertainties in the estimates of the distribution and the magnitude of the BrC climate effect in both remote sensing and modeling (Arola et al., 2011; Feng et al., 2013).

The thermal-optical analysis (TOA) method is one of the most widely used quantitative methods for OC and EC taking use of the difference between the thermo-optical properties of OC and EC (Birch and Cary, 1996; Chow et al., 2004). OC and EC will be volatilized at different heating protocol. The reflectance/transmittance of one laser source (near-infrared wavelength) through the sample filter are continuously monitored and the return of the reflectance/transmittance to its initial value on the thermograph is taken as the split point between OC and EC. This way, the formation of pyrolyzed carbon which can also absorb the light and make the sample darker, is corrected. This method has been wildly used in studies employing the NIOSH protocol or IMPROVE_A protocol (Ji et al., 2016; Chow et al., 2007). However, the thermal-optical approach assumed that EC is the only light-absorbing species, the presence of BrC, which is part of OC but also a light-absorbing component, shifts this separation towards EC, resulting in overestimated EC values and underestimated OC values (Chen et al., 2015; Birch and Cary, 1996).

Sandradewi et al. (2008) pointed out that light absorption measurements at different wavelength by the aethalometer can be used to quantify the contributions of wood combustion and traffic emissions to aerosols since wood smoke contains organic compounds which enhance the light absorption in the ultraviolet wavelength. But traffic emissions produce more black carbon (BC), which dominates the light absorption in the near-infrared wavelength. They took use of aethalometer data measured at 470 nm and 950 nm to quantify the BC distinction between wood

burning and traffic emission. With the similar principle, Wang et al. (2011) used a two-wavelength
Aethalometer (370 and 880 nm) to identify the presence of residential wood combustion (RWC)
particles which was closely associated with BrC. Organic components of wood smoke particles
absorb light at 370 nm more effectively than 880 nm in two-wavelength aethalometer
measurements. They believed that the enhanced absorption (Delta-C=$BC_{370nm}$-$BC_{880nm}$) can serve
as an indicator of RWC particles. This method was further used by Wang et al. (2012a; 2012b).
Chen et al. (2015) used a modified seven-wavelength thermal-optical transmittance/thermal-
optical reflectance (TOT/TOR) instrument (Thermal Spectral Analysis – TSA) allowing the
determination of the OC-EC split at different wavelengths and light absorption measurements to
be made with wavelength-specific loading corrections, providing additional information including
the optical properties of black carbon (BC) and BrC from the IR to UV parts of the solar spectrum
and their contributions. Massabò et al. (2016) further corrected the OC/EC split point using the
Multi-Wavelength Absorbance Analyzer (MWAA) which provides the aerosol absorbance values
at five wavelengths from IR to UV together with a Sunset OC/EC analyzer to achieve the BrC
concentration. With a set of samples collected during wintertime in the Ligurian  Apennines in
Italy, clear correlations were found between the BrC and levoglucosan mass concentration. A
further step of BrC quantification taking use of TSA was reported by Chow et al. (2018), further
proving that the use of seven wavelengths in thermal-optical carbon analysis allows contributions
from biomass burning and secondary organic aerosols to be estimated. Their results clearly
demonstrated the role of BrC in the thermal-optical analysis. However, these techniques focus on
the light absorption measurement of BrC and are still limited reported in previous researches,
though they provide quartz-fiber filter samples that are currently being characterized for OC and
EC by thermal-optical analysis. These methods mentioned above still cannot achieve the
observation of long-term real-time BrC mass concentrations.
Since the establishment of the TOT method by the Sunset Laboratory, the Sunset OC/EC
instrument, as part of the Chemical Speciation Network (CSN), with over 100 monitors across the
United States over 15 years, offering long-term measurement of OC and EC concentrations, has
been widely used in the United States and throughout the world providing important in-situ data
of OC and EC (U.S.EPA, 2019; Birch and Cary, 1996). This instrument had been designed with a
tuned diode laser (red 660 nm) to correct the formation of pyrolyzed carbon. In this study, we
modified the Sunset instrument to a two-wavelength (658 nm and 405 nm) Sunset carbon analyzer
by adding one more violet diode laser at λ=405 nm. The violet diode laser together with the red
diode laser focus through the sample chamber, then the laser beam passes through the filter to
correct for the pyrolysis-induced error. Previous work reported by Chen et al. (2015) as mentioned
above was integrating the optical instrument like the aethalometer to the traditional OC/EC
analyzer; in this way, they provided the light absorption contributions of BC and BrC. The
enhanced carbon analyzer provided new insight into more accurate OC and EC measurements.
Their work was conducted in offline mode; based on their work, our instrument can get the real-
time OC and EC mass concentrations both at 658 nm and 405 nm. BrC particles absorb light at
405 nm more effectively than 658 nm in the two-wavelength Sunset carbon measurements. We
define $dEC=EC_{405\,nm}-EC_{658\,nm}$ and hope it can be an indicator of BrC aerosols so that we can divide
real-time BrC mass concentration measurement from the two-wavelength measurement.
Nanjing, as one of the largest cities in the Yangzi River Delta region, represents a heavy industry
area with a dense population. In addition, due to its topography, Nanjing is very sensitive to
regional transport of air masses from its surrounding areas. OC, EC and dEC were measured from
June 2015 to July 2016 at Nanjing University of Information Science and Technology (NUIST).
Based on the abundant data, together with MODIS fire information, we can analyze the temporal
variation, transport processes and sources of carbonaceous aerosols in North Nanjing and evaluate
the biomass burning impact on dEC, which can be the scientific basis of pollution control policy.
**2. Methods**
**2.1 Study site**
In this study, the sampling site is located at Nanjing University of Information Science and
Technology (NUIST) in the North Suburb of Nanjing (32°207′N, 118°717′E). The study site
is surrounded by housing and industrial areas. Many chemical enterprises, for example, Yangzi
Petrochemical, Nanjing Chemical Industry and Nanjing Iron and Steel Group are located at the
northeast of the study region, which produces exhaust with large amounts of aerosol particles. The
study site is adjacent to a heavily trafficked road (Ningliu Road) located near the site,
approximately 600 m to the east. Therefore, this region has intense human activities, industrial
emissions and heavy traffic flow.
**2.2 Two-wavelength TOT measurement**
Hourly concentrations of OC and EC in $PM_{2.5}$ were sampled and measured by a semi-
continuous carbon analyzer (Model-4, Sunset Lab, USA). Air samples were collected continuously
with a sample flow of ~8 L/min through a $PM_{2.5}$ cyclone. The collection time was set at 45 min
for each cycle. The airstream passed through a parallel plate organic denuder to reduce the effect
of volatile organic compounds and finally deposited on a quartz filter with a diameter of ~17mm.
After a sample was collected, OC and EC were determined using the TOT method by applying
a slightly modified NIOSH 5040 protocol. The details of the heating setup are shown in Table S1.
Figure 1 shows the structure and operational principle of the instrument. Briefly, it consists of two-
stages: the oven is first purged with helium and the oven temperature increased in a stepped ramp
to 840°C, OC is volatilized in this stage. Then the oven temperature is kept at 840°C for a while
and goes down to 550°C. In the second stage, EC is volatilized in a second temperature ramp to
850°C while purging the oven with a mixture containing 2% oxygen and 98% helium. The
pyrolysis products are converted to carbon dioxide ($CO_2$) which is quantified using a self-
contained nondispersive infrared (NDIR) system.
Also, in this study, we used a two-diode lasers (658 nm and 405 nm) equipped Sunset analyzer;
thus mass concentrations of OC and EC at different wavelengths can be measured with the 2-lasers
system. The split point between OC and EC is detected automatically by the RTCalc731 software
provided by Sunset Lab. The principle is the same as for the traditional Sunset carbon analyzer
(Birch and Cary, 1996). An example thermogram of sample analysis using the two-wavelength
Sunset semi-continuous carbon analyzer is shown in Fig. 2. During the sample analysis, the laser
beam at 658 nm and 405 nm are both sent through the filter and the transmitted light signal is
monitored to correct the undesired formation of pyrolyzed carbon (PyrC) and then to determine
the split point of OC and EC at both wavelengths. BrC aerosols absorb light at 405 nm more
significantly than 658 nm in the 2-lasers system. Due to the strong absorption of BrC at the near-
ultraviolet wavelength, the enhanced absorption at 405 nm can serve as an indicator of BrC
aerosols (Liu et al., 2015). We define dEC data as the difference of EC concentrations at two
wavelengths ($dEC=EC_{405nm}-EC_{658nm}$) to identify the presence of BrC aerosols. Our study provides
a one-year measurement of dEC mass concentrations. Besides, OC and EC represent the OC and
EC concentrations at 658 nm in this paper without a special explanation.
At the end of each analysis, a fixed volume of an internal standard containing 5% methane
and 95% Helium is injected and thus a known carbon mass can be derived. The external sucrose
standard (4.207 μg μL$^{-1}$) calibration was conducted every week to insure repeatable quantification.
Calibration with an instrument blank was conducted every day. Both detection limit for OC and

EC of the instrument was 0.5 µg m$^{-3}$. We also did the measurements of OC and EC in PM$_{2.5}$ filter samples using the same method followed by the NIOSH protocol. All data were corrected to blank measurement before comparison. Figure S1 shows the correlations between the real-time OC, EC concentrations and sampling OC, EC concentrations at the same time. The results showed that the online and offline data during the corresponding periods had good correlations with R² of 0.8 for OC, R² of 0.4 for EC and R$^2$ of 0.8 for TC. In order to evaluate the impact of PyrC, we calculated the PyrC at 658 nm fraction of dEC and the average PyrC/dEC was 4.4%, indicating the little influence of PyrC.

**2.3 Test of the new dEC data**

To evaluate the new dEC data, parallel BC concentrations were measured with a seven-wavelength Aethalometer with dEC concentrations in December, 2019. Radiation attenuation of an aerosol deposition on a filter (ATN$_\lambda$) is determined by the Beer-Lambert law:

$$ATN_\lambda = ln\frac{I_{0,\lambda}}{I_\lambda} \qquad (1)$$

Where I$_{0,\lambda}$ and I$_\lambda$ are the measured wavelength-specific laser reflectance signals. ATN$_\lambda$ is used to calculate the attenuation coefficient with Eq. (2):

$$b_{ATN} = \frac{A}{V} \qquad (2)$$

Where A is the filter area and V is the sampled air volume. Then a simplified two-component model is used to calculate the contribution of light attenuation to both BC and BrC (Chow et al., 2018; Chen et al., 2015; Sandradewi et al., 2008; Hareley et al., 2008):

$$b_{ATN}(\lambda) = q_{BC} \times \lambda^{-AAE_{BC}} + q_{BrC} \times \lambda^{-AAE_{BrC}} \qquad (3)$$

Where q$_{BC}$ and q$_{BrC}$ are fitting coefficients, AAE is the absorption Ångström exponent which represents the wavelength-dependent characteristics of light absorption capability of aerosols. The AAE of BC was assumed to be 1. Fitting coefficients in Eq. (3) were obtained for potential AAE$_{BrC}$ between 1 and 8 by least squares linear regression and the AAE$_{BrC}$ leading to the overall best fit in terms of r$^2$ is selected as the effective AAE$_{BrC}$. Using these fitting coefficients, the b$_{ATN}$ due to BC and BrC are calculated at each wavelength. Figure S2 shows that the fitted b$_{ATN}$ at 405 nm are within ±5 % of the measured values for b$_{ATN}$ > 0.01. Figure 3 shows the relationship between the b$_{ATN}$ due to BrC at 405 nm and the dEC. Good correlation between them is found with R square of 0.64, indicating that dEC was associated with BrC.

**2.4 Sampling**

2.4.1 Real-time PM$_{2.5}$ observation
The real-time PM$_{2.5}$ concentrations were measured through the Tapered Element Oscillating
Microbalance (TEOM) method (TEOM1405-DF, Thermo Scientific, America) from August, 2015
to July, 2016. The resolution of the measured data was 6 min. The instrumental operation
maintenance, data assurance and quality control were performed according to the Chinese Ministry
of Environmental Protection Standards for PM$_{10}$ and PM$_{2.5}$ which was named "HJ 653-2013"
(Zhang and Cao, 2015b).
2.4.2 Sample collections
PM$_{2.5}$ in the atmosphere was collected on 8*10 inch prebaked quartz fiber filters (QFF, PALL,
America) by a high volume air sampler (KC-1000, Qingdao, China) at a flow rate of 999 L min$^{-1}$
in four months: 4 June to 18 June, 6 October to 2 November and 10 December to 31 December in
2015, 10 May to 31 May in 2016. Sampling started and ended at around 8:00 and 20:00 every day;
each sample was collected for 12 hours. A total of 148 samples were collected including four field
banks in the four seasons using 10 min exposure to ambient air without active sampling.
All QFFs were pre-baked at 450 °C for 6 h before sampling to remove residual carbon. Before
and after sampling, all QFFs were weighed with an electronic balance (Sartorius, 0.1 mg,
Germany). After weighting, the filters were wrapped in aluminum foils, packed in air-tight
polyethylene bags and stored at -20˚C until further analysis. All procedures during handling of
filters were strictly quality controlled to avoid any possible contamination.
**2.5 Identification of potential regional sources**
The Hybrid Single-Particle Lagrangian Integrated Trajectory (HYSLPIT4.8) model, provided
by the National Oceanic and Atmospheric Administration (NOAA), was used to investigate the air
mass origins of carbonaceous aerosols. The 48-hour back trajectories at Nanjing (32.2°N, 118.7°E)
were calculated every hour (Draxler and Hess, 1998; Rolph et al., 2017; Cohen et al., 2015). In
order to evaluate the behavior of the air mass circulation in the planetary boundary layer (PBL),
the trajectories at 500 m corresponding to the upper-middle height of the PBL were calculated,
representing well-mixed convective boundary layer for regional transport investigation (Xu and
Akhtar, 2010). The National Center for Environmental Prediction Global Data Assimilation
System (NCEP GDAS) data obtained from NOAA with a spatial resolution of 1° ×1° and 24 levels
of the vertical resolution were used as meteorological data input to the model. The Potential Source
Contribution Function (PSCF) model was usually applied to localize the potential sources of
pollutants. The details about the setup of the model can be found in Bao et al. (2017).

**3. Results and discussion**

**3.1 Characteristics of carbonaceous aerosols**

3.1.1 Concentrations of carbonaceous aerosols

The statistics for the $PM_{2.5}$, OC, EC and dEC mass concentrations at the NUIST site are summarized in Table 1. The hourly OC concentrations ranged from 0.5 to 45.8 μg m$^{-3}$ (average of 8.9 ± 5.5 μg m$^{-3}$), and the EC concentrations ranged from 0.0 to 17.6 μg m$^{-3}$ (average of 3.1 ± 2.0 μg m$^{-3}$). The results are comparable to those reported by Chen et al. (2017) in the Xianlin Campus of Nanjing University (5.7 μg m$^{-3}$ for OC and 3.2 μg m$^{-3}$ for EC), which site was located in the southeast suburb of Nanjing and close to the G25 highway and were also affected by traffic sources. The higher OC concentrations in this study are probably due to the around chemical enterprise emissions. The average contributions of OC and EC to the total measured $PM_{2.5}$ mass was 12.8% and 4.3%, respectively, suggesting that carbonaceous fraction made an important contribution to fine particulate matter. The average dEC mass concentration was 0.8 μg m$^{-3}$ contributing 10.0% to OC, 22.3% to EC and 1.3% to the $PM_{2.5}$ concentrations with maximum concentration of 8.1 μg m$^{-3}$ contributing 48.2% to OC, 97.8% to EC and 17.6% to total $PM_{2.5}$ concentrations. This information can be further applied in the PMF analysis to evaluate the sources of the carbonaceous aerosols (Zhu et al., 2014; Sahu et al., 2011; Yan et al., 2019).

Compared with carbonaceous aerosol levels in other cities (Table S2), the OC and EC concentrations in Nanjing were generally lower than those observed in urban sites such as Beijing and Shanghai and inland cities like Chengdu and Chongqing which are affected by the basin terrain characteristics with static wind and unfavorable diffusion conditions, but higher than those observed in the southern coastal cities such as Guangzhou, which is a megacity in China. It could be explained since the site in Guangzhou was a rural site. In general, the level of carbonaceous aerosol concentrations in China is higher than that in developed countries in the United States and Europe and lower than that in developing countries like India, though the sampling period in India was from late autumn to winter and the much higher concentrations in India indicated the heavy pollution level. The average OC/EC ratio in this study was 3.6, which is lower than most of those reported in other studies, indicating the important impact of vehicle emissions at our study site.

Figure 4 shows the mass fractions of hourly carbonaceous aerosols and OC/EC ratios at different $PM_{2.5}$ concentration intervals during the study period. During that period, 84.2% of the

PM$_{2.5}$ samples exceeded the daily averaged Chinese national ambient air quality standard (NAAQS)
of 35.0 μg m$^{-3}$ for the first grade and 40.1% of the total samples exceeded the NAAQS of 75.0 μg
m$^{-3}$ for the second grade, reflecting heavy aerosol pollution in the study area. Generally, the
fractions of carbonaceous components decreased with increasing PM$_{2.5}$ pollution level. A larger
mass fraction (about 32.3%) of carbonaceous aerosols in PM$_{2.5}$ was found for relatively lower
PM$_{2.5}$ levels (0–20 μg m$^{-3}$) compared to high PM$_{2.5}$ levels (300–500 μg m$^{-3}$) with a carbonaceous
aerosol mass fraction of 5.2%. The results indicate that other components like secondary inorganic
aerosol (SIA) contribute more significantly to heavy haze events in Nanjing, which was also found
in other cities in the Yangtze River Delta area (Yang et al., 2011; Zhang and Zhang, 2019). The
contribution of dEC to OC decreased with the increase of PM$_{2.5}$ concentrations between 0-200 μg
m$^{-3}$, and then increased with the increase of PM$_{2.5}$ concentrations between 200-500 μg m$^{-3}$. The
dEC contributed most significantly to OC of 14.3% for PM$_{2.5}$ concentrations below 20 μg m$^{-3}$. A
similar trend was found for the OC/EC ratios which showed a sharp increase along with enhanced
PM$_{2.5}$ level above 150 μg m$^{-3}$. Previous studies have reported that high OC/EC ratios were related
to SOC formation or biomass burning emissions whereas low OC/EC ratios were related to vehicle
exhaust (Wang et al., 2015). We divided the dEC/OC at different intervals of OC/EC ratios and
found that the dEC/OC increased when the OC/EC ratios increased in the four seasons, indicating
strong secondary sources or biomass burning contributions to dEC during heavy pollution periods
(Fig. S3).
3.1.2 Seasonal variations of carbonaceous aerosols
As shown in Fig. 5, the OC, EC, dEC concentrations and dEC/OC ratios showed similar
variations with highest in winter and lowest in summer. The average OC and EC concentration in
winter was ~1.4 times and 1.5 times higher than that in summer and the average dEC
concentrations and dEC/OC in winter were approximately 1.4 and 1.6 times higher than those in
summer (Table 1). High dEC/OC was found in January and February in winter, indicating strong
influence of anthropogenic sources on dEC, such as coal combustion. In addition, we found strong
biomass burning activities in February, which significantly contributed to the high concentrations
of dEC in February; more details can be found in section 3.3. The seasonality of carbonaceous
species in PM$_{2.5}$ was strongly influenced by seasonal variations in emission intensities and
meteorological parameters. Table S3 summarizes the meteorological parameters in the four
seasons during the study period. The high carbonaceous aerosol concentrations in winter were

mainly a result of relatively stable atmospheric conditions with low temperature, relative humidity and boundary layer on one hand, and on the other hand, increasing emissions from fossil-fuel combustion for heating from the chemical enterprises nearby. In summer, the higher boundary layer resulted in the dispersion of aerosols in the atmosphere, and the higher temperature promoted the partitioning of semi-volatile organic compounds (SVOCs) into the gaseous phase (Yang et al., 2011). In addition, large precipitation in summer (586 mm in total) favored the wet scavenging processes of aerosols.

The average OC/EC ratios in spring, summer, autumn and winter were 3.9, 4.0, 2.8 and 3.4, respectively (Table 1). The OC/EC ratio could give some information about primary and secondary organic carbon (Turpin and Huntzicker, 1995; Lim and Turpin, 2002). In summer, strong convective activities in the atmospheric boundary layer and solar radiation, high temperature and plenty of moisture in the atmosphere were favorable for the formation of SOC. On the other hand, the high OC/EC ratios in June in this study were also strongly related to biomass burning which will be discussed in 3.3 sections. The lower ratios of OC to EC in autumn and winter indicate strong primary sources in these two seasons. It should be noted that the OC/EC ratios were a rough indicator to estimate the primary and secondary organic carbon; further analysis of the formation of SOC needs to be conducted in the future (Pio et al., 2011; Wu and Yu, 2016).

### 3.1.3 Diurnal variation of carbonaceous aerosols

The diurnal pattern of carbonaceous aerosols can be affected by both meteorological parameters and sources (Ji et al., 2016). Figure 6 depicts the diurnal variation of OC, EC, dEC, dEC/OC and OC/EC ratios during the study period. Clear diurnal variations were observed in OC and EC. Both the OC and EC concentrations kept high levels at night and low levels in the daytime, indicating the strong influence of the atmospheric boundary layer on air quality in northern Nanjing. The peak occurred in the morning both for OC and EC, indicating the significant impact of traffic sources on the OC and EC concentrations. The dEC/OC and OC/EC ratios showed similar trends in the daytime with gradually increase from morning till afternoon, indicating the importance of the contribution of secondary sources to dEC. Similar though not so obvious diurnal variations were found in dEC. It should be noted that the vehicle emissions and the boundary layer height had no significant effect on the diurnal variation of dEC/OC, suggesting there were no significant local sources of dEC. There was a small peak in dEC/OC at 3:00 am, which might be related to the aqueous secondary organic aerosol formation during nighttime (Sullivan et al., 2016).

The relative humidity (RH) and Temperature (T) dependent distributions of OC, EC mass
concentrations and dEC/OC and OC/EC throughout the study period are shown in Fig. 7. High
dEC/OC (>30 %) can be found in three areas, first shown in the right area with relatively high T
at 25-40 °C and RH at 40-60 %, which were usually found in the summer afternoon which was
closely related to the strong formation of SOC. This distribution was also seen in OC/EC. The
second area is in the upper region with RH over 80 % and T at 10-20 °C and the third area appears
for RH below 30 % and T at about 10 °C, corresponding to nighttime and winter afternoon. In
general, dEC had no strong dependence on the RH and T distribution, indicating the complex
formation mechanism of dEC. OC and EC show similar distributions with the highest mass loading
(OC: $> 20$ $\mu g$ $m^{-3}$; EC: $> 8$ $\mu g$ $m^{-3}$) at relatively high RH at 60-80 % which usually occurred at
night with relatively low boundary layer height, leading to the accumulation of aerosols. However,
the corresponding OC/EC ratios were low, suggesting the importance of primary sources to OC
and EC in northern Nanjing, which will be verified in the wind rose of OC and EC (Fig. 8).
**3.2 Air mass transport**
3.2.1 Windrose of carbonaceous aerosols
To investigate the influence of air mass transport to the study site, the wind rose of OC, EC and
dEC/OC using hourly data in the four seasons is shown in Fig. 8 (Carslaw and Ropkins, 2012).
Two points should be noted. First, high OC and EC mass concentrations were found near the field
site (indicated by wind speed (WS) $< 1$ $m$ $s^{-1}$), suggesting that local and primary emissions (e.g.,
industrial and vehicle emissions) were stable and important sources contributing to atmospheric
OC and EC mass concentrations in northern Nanjing. The OC mass concentrations from the
southwest increased with the increase of WS in summer, indicating that the sources of OC are
complicated in summer including secondary reaction during long-range or regional transport.
Second, compared with OC and EC, dEC showed no significant local sources. The dEC/OC
increased with increasing WS and the highest dEC/OC were found for WS over 3 $m$ $s^{-1}$. Long-
range or regional transport were highly likely the main sources contributing to the dEC mass
concentrations.
3.2.2 The potential source areas of carbonaceous aerosols
The possible source contributions were evaluated using the PSCF model and the PSCF maps
are shown in Fig. 9 (Petit et al., 2017). The areas with high PSCF values were highly likely the
potential pollution source areas. As shown in Fig. 9, the PSCF results further proved the strong

regional transport contribution to dEC and local contributions to OC and EC. In spring, the potential source areas of OC and EC were mainly from the southwest of Nanjing; however, the potential source areas of dEC were from the east of Nanjing, indicating obvious different sources between OC, EC and dEC. In summer, local areas were the main source areas of EC and the near-by Yangtze River Delta City Group from the southeast of Nanjing including developed cities like Shanghai were the main sources areas of OC and dEC. The anthropogenic emissions from these areas might be important sources of OC and dEC. Besides, both the potential sources areas of dEC and EC were in the northwest of Nanjing in summer, suggesting strong primary sources of dEC from this area which were very likely associated with biomass burning, more details are given in section 3.3. In autumn, local sources from the study site were strongest for OC and EC. However, dEC mainly originated from regional transport from the northwest and southeast areas of Nanjing. Biomass burning has been proved to be an important source of air pollutants in the Yangtze River Delta (YRD) area, especially in the wheat harvest seasons (e.g., June and October) (Cheng et al., 2014; Zhang and Cao, 2015a). In addition, the YRD area is the most economically developed region in China and has lots of industrial cities, which means that industrial emissions and anthropogenic sources contributed to high carbonaceous aerosol pollution levels. In winter, dEC was mainly from long-range transport from northern cities and regional transport from the southwest areas of Nanjing while both long-range transport and local sources were found in OC and EC concentrations.

**3.3 The characteristics of carbonaceous aerosols during biomass burning periods**

The biomass burning emission has been proved to be an important source of BrC on a global scale; it is consistently observed in large-scale forest fire events (Laskin et al., 2015). Based on the Fire Information for Resource Management System (FIRMS) derived from the Moderate Resolution Imaging Spectroradiometer (MODIS), we found that the fire points amounted to 2028, 1773 and 967 on 11 Jun 2015, 7 February 2016 and 2 March 2016, respectively, in the areas around our study site, suggesting there were strong biomass burning events on these days (Fig. S4). To further investigate the biomass burning impact on dEC, we analyzed the temporal trends of carbonaceous aerosols from 4 June 2015 to 19 June 2015 and 7 February 2016 to 3 March 2016, respectively. Combining the observed aerosol concentrations and fire information, we divided the periods into normal days and biomass burning days. It should be noted that the biomass burning days are not determined based only on fire points. We also considered the 48-h backward

trajectories and open biomass burning areas. For example, we found lots of fire points from 11
June 2015 to 12 June 2015 and from 7 February 2016 to 10 February 2016, respectively, and the
48-h back trajectories passed over these biomass burning areas (Fig. S5b, c). However, although
there were large amounts of fire points in the northwest of Nanjing from 8 June 2015 to 9 June
2015, the backward trajectories showed that the air masses during the periods came from the
southeast areas where no open fire points were found (Fig. S5a). In contrast, there were only a few
fire points found near the study site from 26 February 2016 to 27 February 2016, the 48-h backward
trajectories showed the air masses came exactly from the area (Fig. S5d).
As shown in Fig. 10 and Fig. 11, we found that dEC concentrations, dEC/OC and OC/EC
ratios showed peaks during each biomass burning period which were not that obvious in OC and
EC concentrations, suggesting the unique biomass burning impact on dEC and the sources of OC
and EC were more complicated. It should be noted that there were peaks of dEC on 9 June 2015
and 13 February 2016, which were not biomass burning days, suggesting that biomass burning
was not the only source of dEC. As mentioned in sections 3.1 and 3.2, anthropogenic emissions
could be the sources of dEC and the secondary sources cannot be ignored, either. Summarized in
Table 2 are the average and standard deviation values of OC, EC, OC/EC, dEC and dEC/OC during
biomass burning and normal days. The OC/EC, dEC concentrations and dEC/OC were obviously
higher in biomass burning days than in normal days, but similar levels of the OC and EC
concentrations were found both in biomass burning days and normal days in summer, suggesting
the great contribution of biomass burning emissions to dEC and there were other sources of OC
and EC in summer. All the carbonaceous aerosols were higher in biomass burning days in winter;
in addition, the locations of open fire counts were mainly in the northwest and southwest area of
the study site (Fig. S5c, d), which were the potential source areas of OC, EC and dEC in winter as
discussed in section 3.2.2, indicating strong contributions of biomass burning emissions to all the
carbonaceous aerosols in winter.
**4. Conclusions**
In this study, the characteristics and sources of carbonaceous aerosols in North Nanjing were
investigated and we introduced a two-wavelength method by modifying the Sunset carbon analyzer.
We incorporated a new diode laser at $\lambda=405$ nm in the instrument, making it possible to detect the
laser beam passing through the filter at both wavelength $\lambda=658$ nm and $\lambda=405$ nm, so that we can
obtain the dEC concentrations. Our study illustrated the feasibility of using dEC to characterize
the BrC aerosols, providing a new idea about the measurement of BrC. The results showed that
high (low) OC, EC and dEC concentrations were found in winter (summer), indicating the
significant impact of the increase of various emission sources in winter and wet scavenging by rain
in summer. Similar diurnal cycles for OC and EC concentrations were found with high at night
and low in daytime, strongly affected by the boundary layer. Traffic emissions were found to have
a significant influence on the concentrations of OC and EC. Similar trends were found in the
diurnal cycle of dEC/OC and OC/EC and the dEC/OC increased when the OC/EC ratio increased,
indicating strong secondary sources or biomass burning impact on dEC. The wind rose and
receptor model results showed that strong local emissions were found for OC and EC; however,
dEC was significantly affected by regional or long-range transport. The near-by YRD area was one
of the main potential source areas of dEC, suggesting that anthropogenic emissions could be the
sources of dEC. Together with the back trajectories analysis and MODIS fire information, we
analyzed two biomass burning events both in summer and winter. The results showed that the
sources of OC and EC were more complicated than those of dEC in summer. Biomass burning
emission made a great contribution to dEC concentrations in summer. A large number of open fire
counts from the northwest and southwest areas of the study site was observed; these fires
significantly contributed to the carbonaceous aerosol pollution.
Our modified two-wavelength instrument provided more information than the traditional
single-wavelength thermal-optical carbon analyzer. The results proved that dEC can be an
indicator of BrC in biomass burning days. It should be noted that the sources of dEC were
complicated and the anthropogenic emissions and secondary formations of dEC aerosols could not
be ignored; further chemical analysis needs to be conducted in the future. The evaluation of SOC
formation and the relationship between dEC and SOC can be conducted. In addition, more
chemical analysis such as the analysis for ions, organic matter or sugars in $PM_{2.5}$ can be made;
thus we can get some information of the tracers of different sources and more accurate and
quantitative source apportionment can be done (Bhattaraia et al., 2019; Wu et al., 2018; 2019). We
also hope that the dEC data can be further applied in more research.

**Acknowledgments**
This research was financially supported by the National Natural Science Foundation of China
(grant no. 41977305), the Provincial Natural Science Foundation of Jiangsu (grant no.

BK20180040) and the Postgraduate Research & Practice Innovation Program of Jiangsu Province (grant no. KYCX18_1014). This study was supported by the funding of Jiangsu Innovation & Entrepreneurship Team. The authors would also like to thank the China Scholarship Council for the support to Mengying Bao. We would also like to express our gratitude to Yuanyuan Zhang, Zufei Xu and Tianran Zhang for their assistance in the instrument maintenance throughout the observation period. Besides, we are grateful for Prof. Yunhua Chang, who made considerable comments and suggestions for this paper.

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

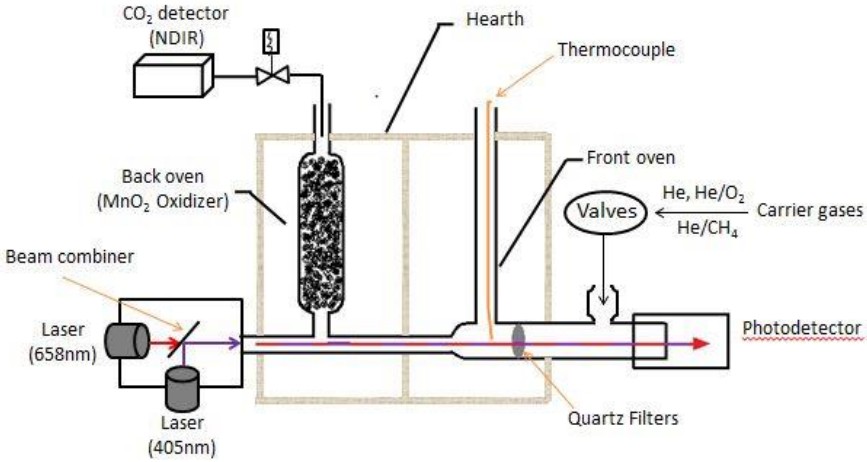


**Figure 1**. Principle and structure of the Sunset semi-continuous carbon analyzer.

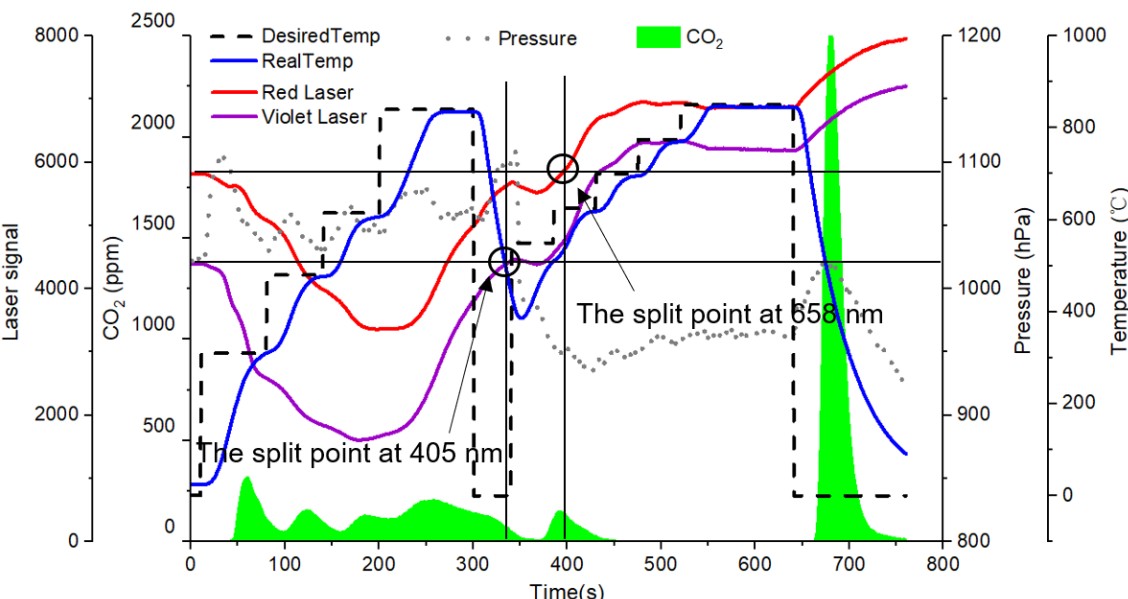


**Figure 2.** Example thermogram of sample analysis using the two-wavelength Sunset semi-
continuous carbon analyzer.

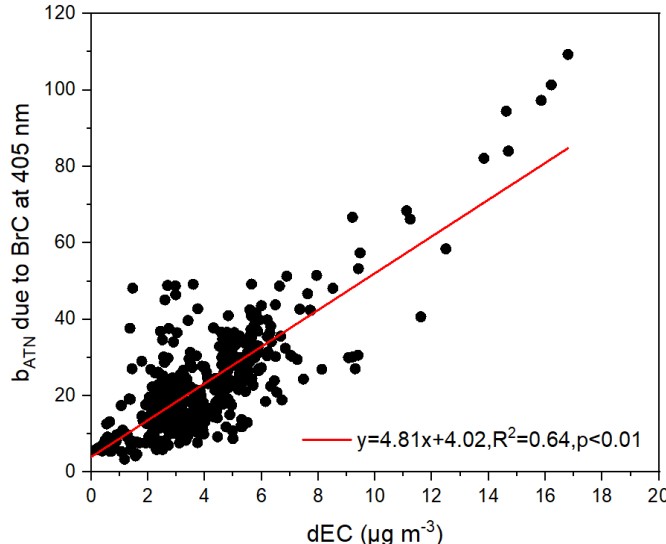


**Figure 3.** Relationship between the $b_{ATN}$ due to BrC at 405 nm and the dEC concentrations.

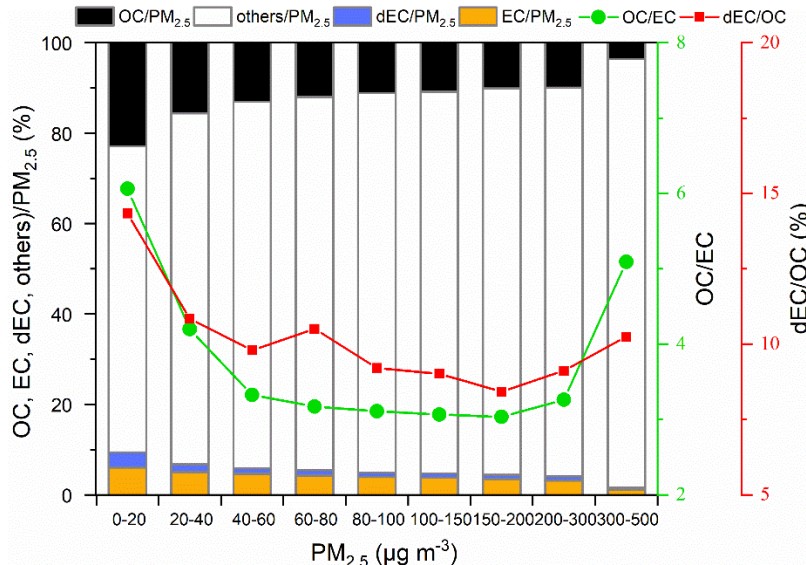

**Figure** **4**. Carbonaceous species fractions of PM$_{2.5}$ and OC/EC ratios at different PM$_{2.5}$
concentration intervals at NUIST from June 2015 to August 2016.

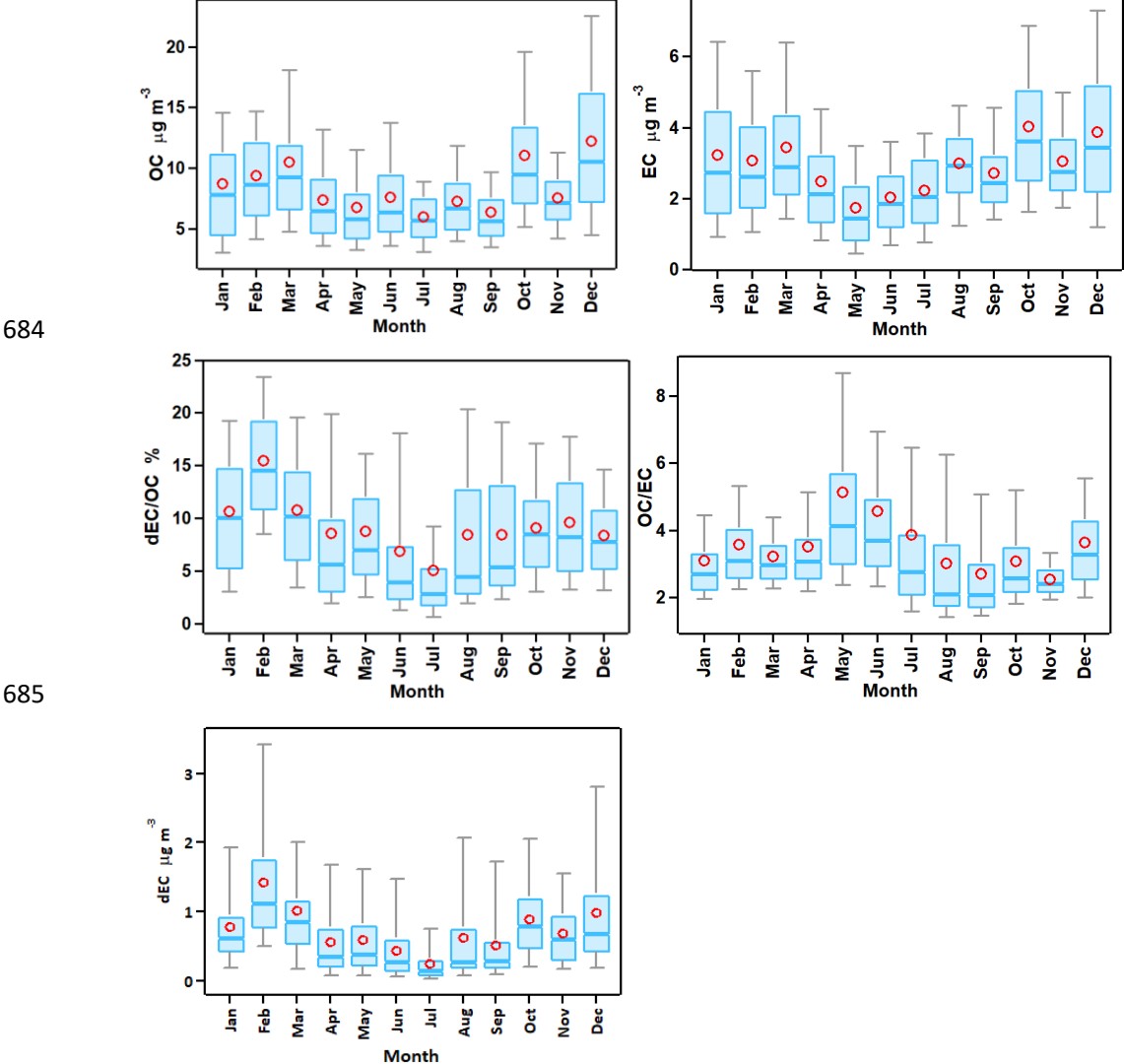



**Figure 5.** Monthly variations of OC, EC, dEC, dEC/OC and OC/EC ratios at NUIST from June
2015 to August 2016. The boundary of the box indicates the 25% and 75% percentile,
respectively. The lower and upper whiskers indicate the 10% and 90% percentile, respectively.
The red circle within the box marks the average while the line within the box marks the median.

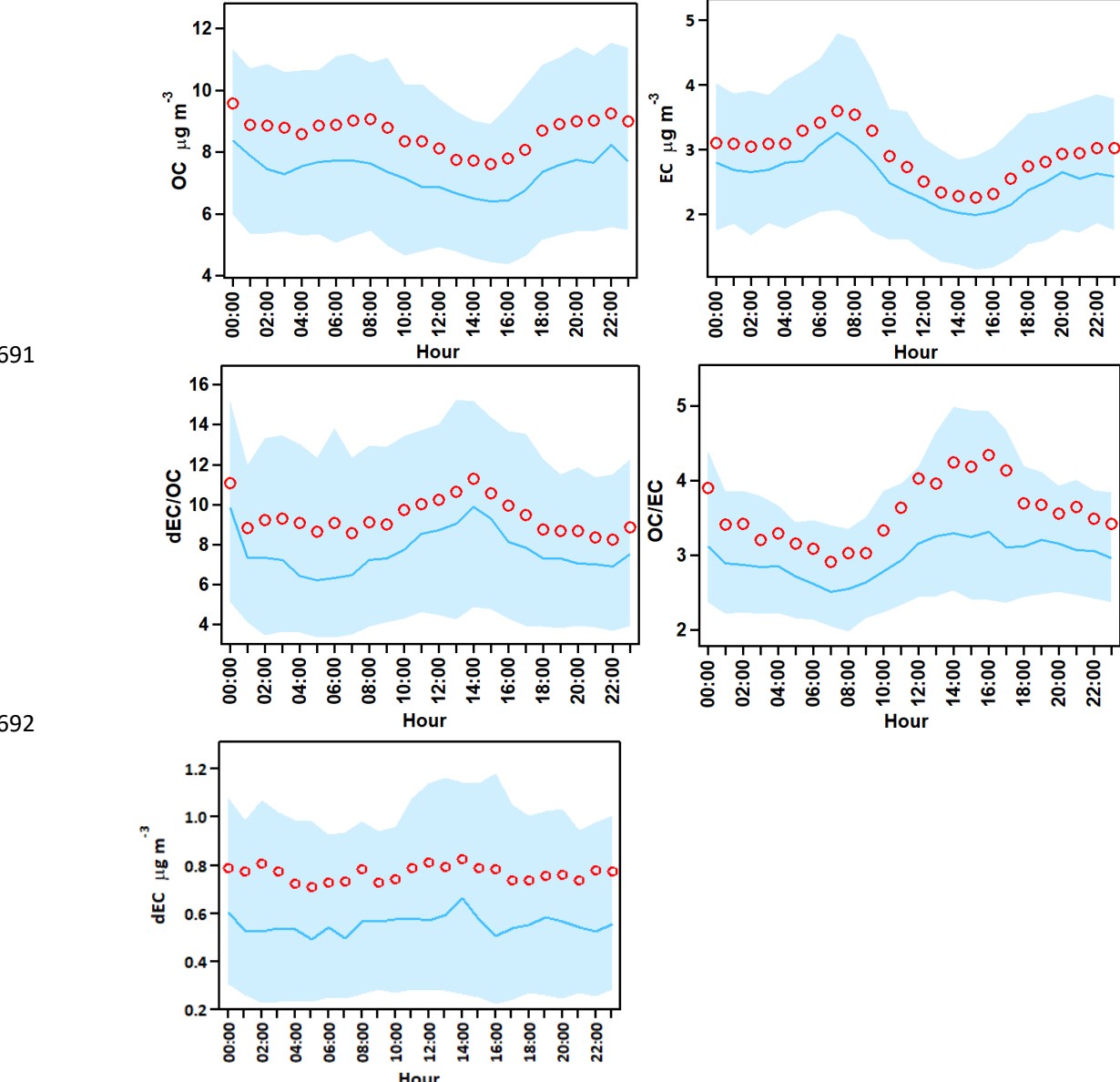


**Figure 6.** Diurnal variations of OC, EC, dEC concentrations, dEC/OC and OC/EC ratios during
the study period. The boundary of the shaded area indicates the 25% and 75% percentile,
respectively. The red circle marks the average while the blue line marks the median.

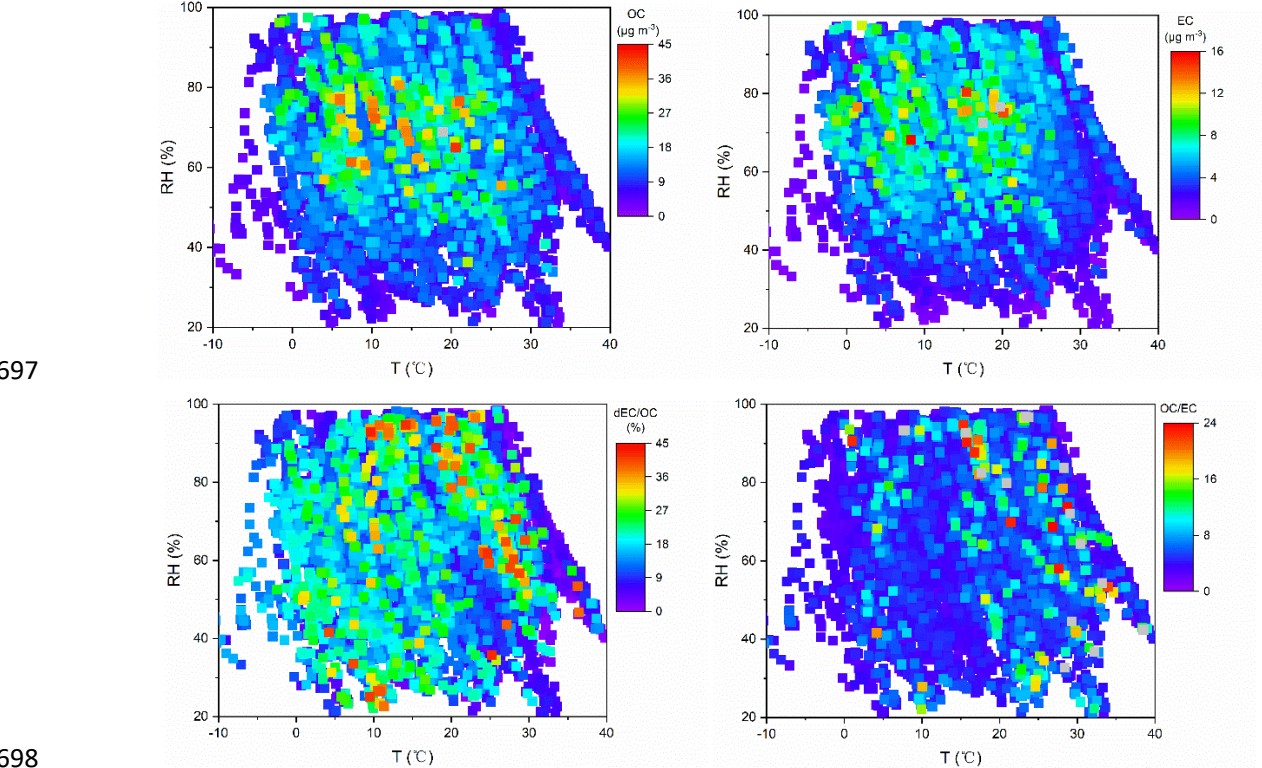


**Figure 7.** RH/T dependence of OC, EC, dEC/OC and OC/EC ratios during the study period.

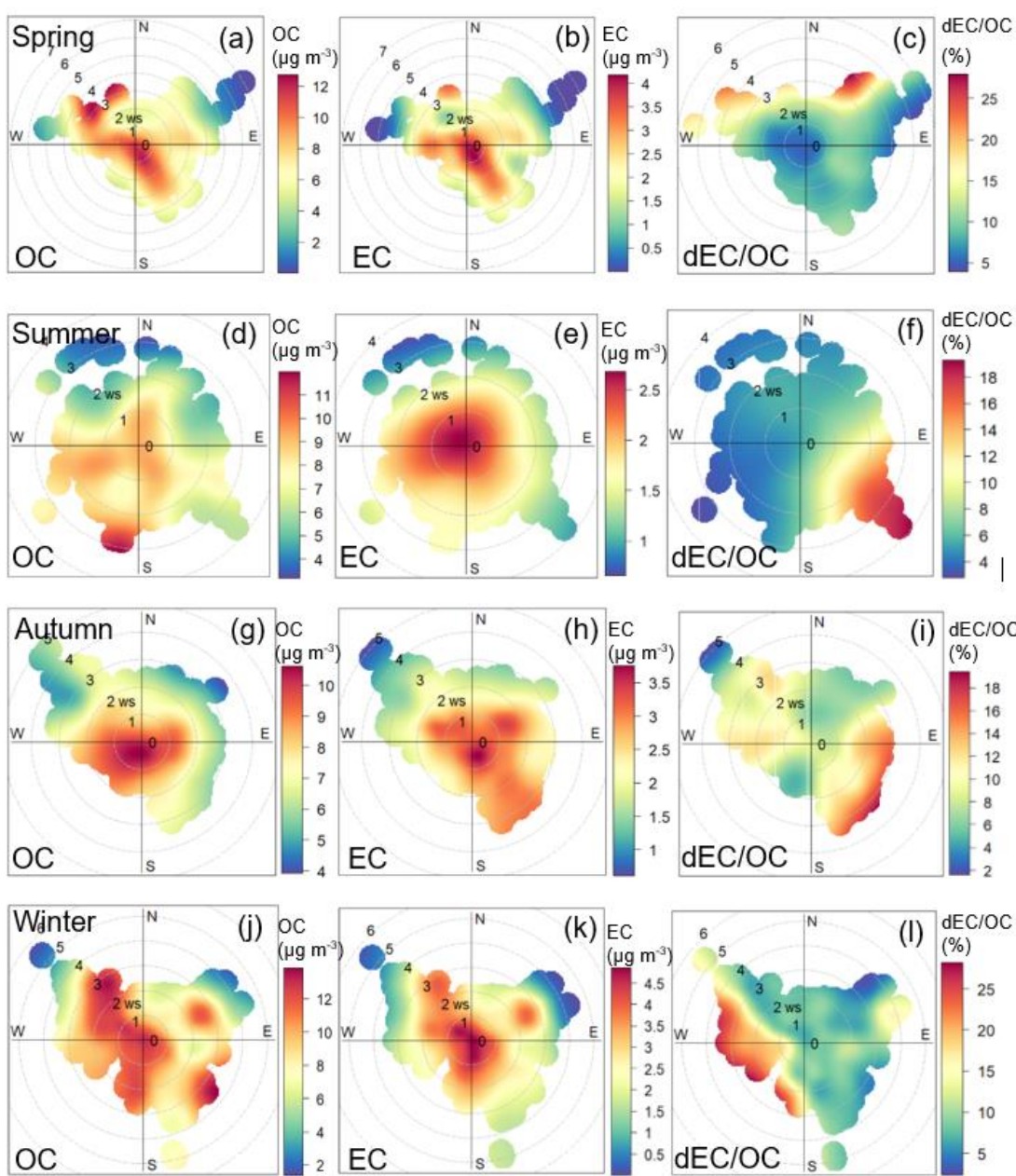

**Figure 8.** Wind rose of OC, EC and dEC/OC in spring ((a), (b), (c)), summer ((d), (e), (f)), autumn
(g), (h), (i)) and winter ((j), (k), (l)).

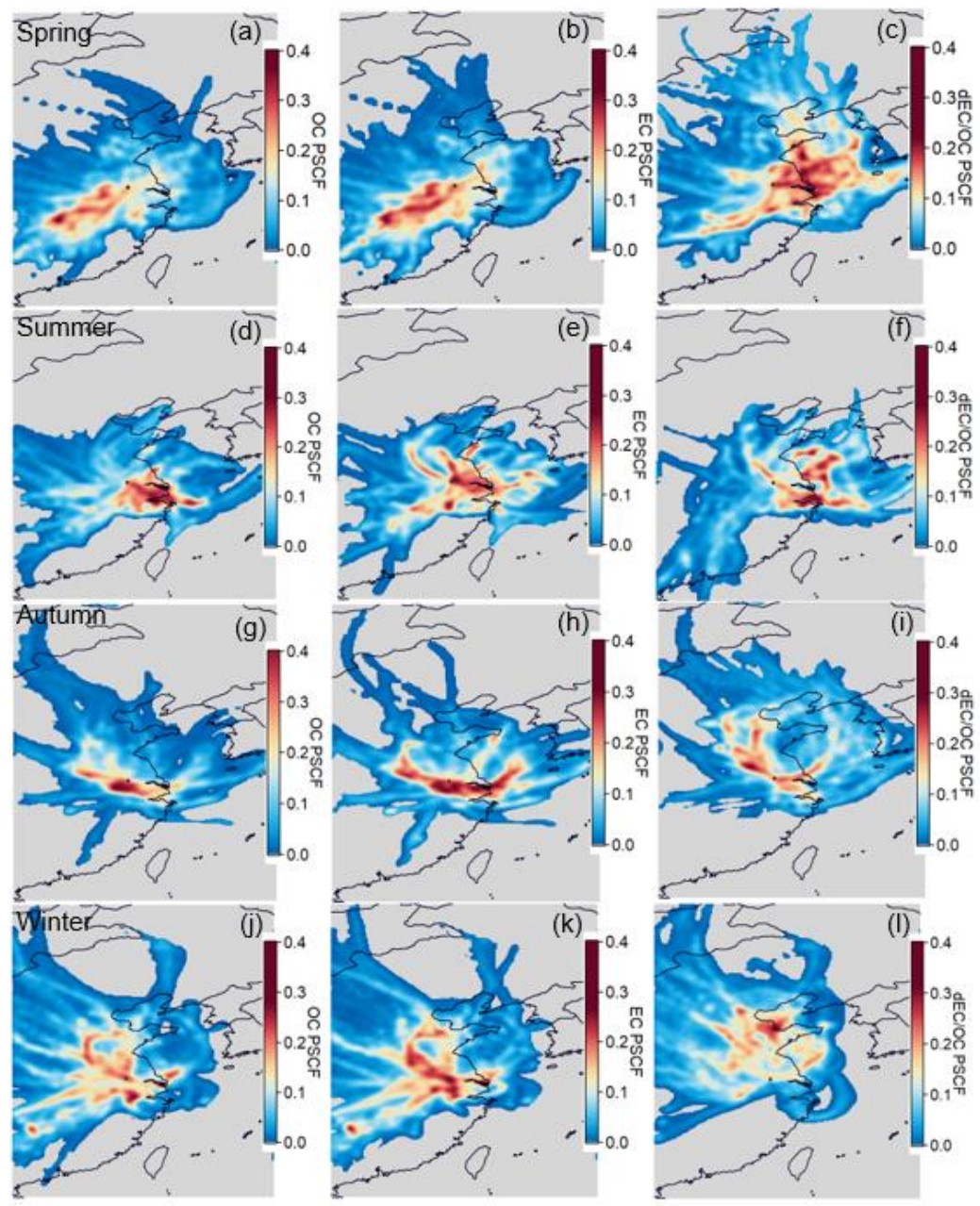

**Figure 9.** PSCF map for OC, EC and dEC/OC in spring ((a), (b), (c)), summer ((d), (e), (f)), autumn
(g), (h), (i)) and winter ((j), (k), (l)).

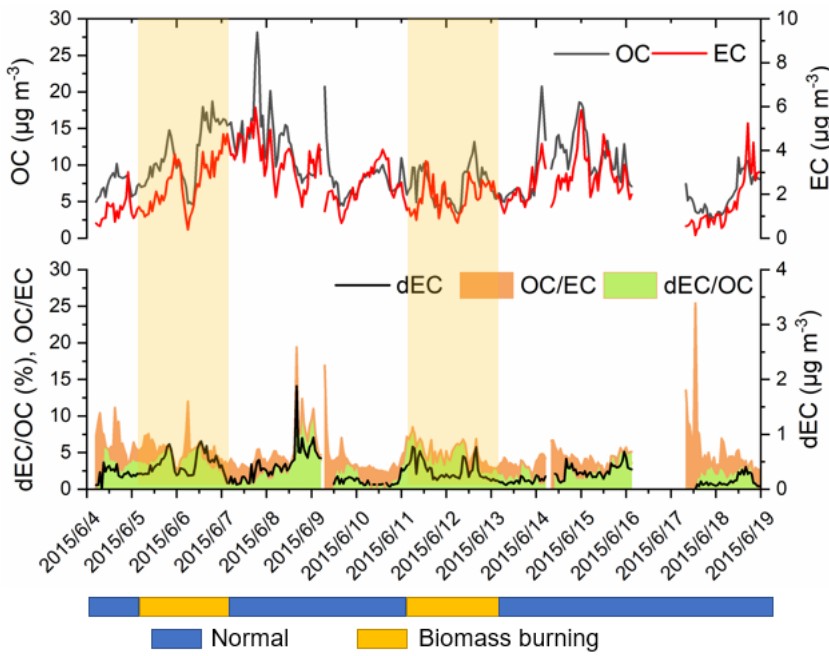

**Figure 10.** Time series of OC, EC, dEC/OC, dEC and OC/ EC from 4 June 2015 to 19 June 2015.
The period was divided into normal days (blue bars) and biomass burning days (yellow bars). The
yellow shadow represents the biomass burning periods.

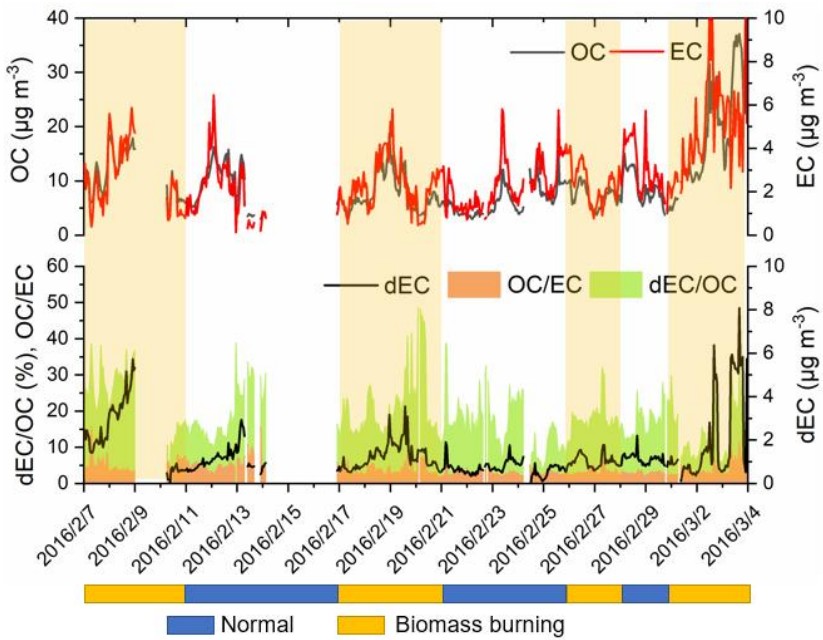

**Figure 11.** Time series of OC, EC, dEC/OC, dEC and OC/ EC from 7 February 2016 to 3 March
2016. The period was divided into normal days (blue bars) and biomass burning days (yellow bars).
The yellow shadow represents the biomass burning periods.
**Table 1.** Statistical summary on the PM$_{2.5}$ and carbon species concentrations.

| N=5113 | Annual | | | | | Spring | Summer | Autumn | winter |
|---|---|---|---|---|---|---|---|---|---|
| | Average | Standard Deviation | Median | Min | Max | Average | Average | Average | Average |
| PM$_{2.5}$ (µg m$^{-3}$) | 77.2 | 48.6 | 65.0 | 2.5 | 458.1 | 72.1 | 47.9 | 70.5 | 91.8 |
| OC (µg m$^{-3}$) | 8.9 | 5.5 | 7.5 | 0.5 | 45.8 | 8.4 | 7.2 | 8.4 | 10.2 |
| EC (µg m$^{-3}$) | 3.1 | 2.0 | 2.6 | 0.0 | 17.6 | 2.6 | 2.3 | 3.3 | 3.4 |
| OC/EC | 3.5 | 2.4 | 2.9 | 1.0 | 29.3 | 3.9 | 4.0 | 2.8 | 3.4 |
| dEC (µg m$^{-3}$) | 0.8 | 0.8 | 0.6 | 0.0 | 8.1 | 0.8 | 0.5 | 0.7 | 1.1 |
| dEC/OC (%) | 10.0 | 7.2 | 8.6 | 0.0 | 48.2 | 9.5 | 6.9 | 9.0 | 11.3 |
| dEC/EC (%) | 22.3 | 16.7 | 18.5 | 0.1 | 97.8 | 24.5 | 18.2 | 18.7 | 25.9 |
| OC/PM$_{2.5}$ (%) | 12.8 | 5.6 | 11.6 | 0.7 | 66.2 | 13.2 | 14.4 | 14.1 | 11.1 |
| EC/PM$_{2.5}$ (%) | 4.3 | 2.3 | 3.9 | 0.0 | 33.2 | 3.9 | 4.7 | 5.8 | 3.7 |
| dEC/PM$_{2.5}$ (%) | 1.3 | 1.2 | 0.9 | 0.0 | 17.6 | 1.4 | 1.3 | 1.2 | 1.3 |


**Table 2.** Statistics of OC, EC, OC/EC, dEC and dEC/OC during biomass burning days and normal
days. The values represent average±standard deviation.

| | | OC ($\mu g\ m^{-3}$) | EC ($\mu g\ m^{-3}$) | OC/EC | dEC ($\mu g\ m^{-3}$) | dEC/OC (%) |
|---|---|---|---|---|---|---|
| June 4th to 19th | Normal days | 9.5±4.5 | 2.6±1.3 | 4.3±2.3 | 0.2±0.1 | 2.5±1.3 |
| | Biomass burning days | 9.0±3.6 | 2.0±0.9 | 4.8±1.6 | 0.4±0.2 | 4.6±1.4 |
| February 7th to March 3rd | Normal days | 7.5±3.3 | 2.5±1.2 | 3.3±1.3 | 0.8±0.3 | 12.7±5.6 |
| | Biomass burning days | 11.2±7.2 | 3.1±1.9 | 4.0±1.8 | 1.7±1.4 | 15.4±7.8 |
