# Peer review of "Highly time-resolved characterization of carbonaceous aerosols using a two-wavelength"

_Atmospheric Measurement Techniques, 2020_

## Referee Comment (RC1) · Anonymous Referee #2 · 20 Nov 2020

Manuscript by Bao et al. describes the annual measurement of carbonaceous aerosols in Nanjing, China, using a two-wavelength Sunset Lab. semi-online analyzer. If I understand well, this is the first article where a semi-online Sunset EC/OC analyzer with a dual laser is presented. The manuscript thus shows both an improved instrument for measuring of EC/OC and a demonstration of the newly measured value dEC, which is presented as an alternative indicator to brown carbon (BrC) measurement. From this point of view, the publication in AMT is logical. However, I have a few comments and recommendations.

General comments:

- Since this is the first described use of a two-wavelength analyzer, the article should contain more information about instrument itself. It means how it differs from previous versions of the instrument, what software is used to evaluate data, and authors also should provide a Figure with an example of a typical analysis. What are the differences, for example, in comparison with the 7-wave offline DRI analyzer (Chen et al., 2015). It is also not clear from the description in the methodology how the split between OC and EC is determined. Are there any corrections used?

- How much of the dEC roughly overlaps with the pyrolyzed carbon (PyrC) that was determined at the EC658nm split? This information can be very useful for previous studies where the PyrC level has been reported.

- What was the distribution of dEC within the EC fractions? This information would be particularly valuable for studies in which OC and EC fractions are used for PMF analyzes in determining sources (e.g., Sahu et al., 2011; Yan et al., 2019; Zhu et al., 2014).

- Many studies compare the relationship between BC and EC (e.g., Jeong et al., 2004; Karanasiou et al., 2020) to determine the relationship between quantities that are determined by different measurement methods. While BC reflects the optical properties of the aerosol, EC rather tends to define the chemical composition. However, the relationship between them undoubtedly exists (Petzold et al., 2013). An analogous relationship should be between the optically determined BrC and the thermo-optically determined dEC, which is newly defined by the authors of this manuscript. In the introduction to the manuscript, the authors outline relationship between BrC a dEC, but further in the text, they continue with this statement only as not proved hypothesis that dEC is an alternative to BrC. However, this relationship should be demonstrated from parallel BC data measured optically at different wavelengths, e.g., from aethalometer (Sandradewi et al., 2008). If the authors do not have parallel measured data from the aethalometer, then optically measured BC data can be obtained directly from the EC/OC analyzer measurements (Chen et al., 2015; Vodicka et al., 2020; Ziková et al.,

2016). The subsequent determination of BrC can then be performed similarly to Chow et al. (2018).

Specific comments:

lines 82-85: "This method has been wildly used in present studies applied in the thermal–optical transmittance (TOT) Sunset carbon analyzer based on NIOSH protocol or thermal–optical reflectance (TOR) Desert Research Institute (DRI) carbon analyzer based on IMPROVE_A protocol (Ji et al., 2016)" First, rather "widely" than "wildly". Second, the sentence is not completely correct because it is possible to use both NIOSH and IMPROVE or another temperature protocol on both devices from Sunset or DRI. Third, in a referenced paper by Ji et al. (2016), there is nothing about the IMPROVE_A protocol. About IMPROVE_A protocol is a paper by Chow et al. (2007).

line 110: "Wang et al. (2011) used a two-wavelength Aethalometer...." The original work in which this type of BC distinction between wood burning and traffic emission was used is Sandradewi et al. (2008), which should be noted. Further, a reference to the work of Wang et al. (2011) is not in references.

line 102: "Italian Apennines" are quite broad area. Massabó specifically states the Ligurian Apennines, Italy. By the way, reference to the work of Massabó et al. (2016) disappeared from references in second version of manuscript.

line 116: "important situ data" change to "important in-situ data"

line 148: "January-April in 2017" Why the authors report periods that they do not use in the evaluated data?

line 153: ". . .collected on prebaked quartz fiber filters. . ." Indicate temperature and time of filter prebaking.

line 170: "of ∼ 17mm" Filter diameter in Model-4 from Sunset Lab. is usually 16 mm. Is it different in the new type of device with two lasers?

line 172: "modified NIOSH 5040 protocol" If the used NIOSH protocol was modified, the authors should describe it in detail (step, time, temperature) or give a reference where the protocol is described. In addition, the authors should explain why they chose NIOSH protocol which usually underestimate the EC (e.g., Bautista VII et al., 2015; Chow et al., 2001).

line 193: "We also did the measurements of OC and EC in PM 2.5 filter samples using the same method followed by the NIOSH protocol." Were offline measurements also performed on a two-wavelength analyzer? If yes, please provide a comparison of the dEC. Second, were both data sets corrected (or uncorrected) to blank measurements before comparison?

line 202: "(Draxler and Hess, 1998)" Authors of HYSPLIT prefer newer citations to their model - see: https://www.ready.noaa.gov/HYSPLIT_traj.php

line 233: "The average OC/EC ratios in this study was 3.6, which was lower than most of those reported in other studies..." This ratio depends, among other things, on the protocol used. If you are comparing with other studies, they need to use also the NIOSH protocol. If you compare with studies where they use, for example, the IMPROVE protocol, which generally analyzes higher concentrations of EC than the NIOSH protocol, then the OC/EC ratio between these studies is different.

lines 271-273: "The OC/EC ratio could give some information about primary and secondary organic carbon (Turpin and Huntzicker, 1995; Lim and Turpin, 2002)" The OC/EC ratio can be a rough indicator of the presence of primary and secondary carbon aerosols. Usually, this analysis is based on the determination of the OC/ECpri ratio and is applicable under certain conditions (Pio et al., 2011). If the authors want to discuss the share of primary and secondary OCs, they should use some more recent approach – see, e.g., Wu and Yu (2016).

line 294: "...shown in Fig. 5. We also found similar distributions in dEC/OC and OC/EC." Not clear what authors mean by similar distribution in dEC/OC and OC/EC. In Fig.

5, color maps of these two parameters are different. Moreover, the conclusions that authors draw from it are not very clear from Figure 5. Maybe some other depiction of temperature and RH dependence would be more appropriate.

line 303: Replace "aass" by "mass".

line 323: Replace "Local" by "local".

Table 1: Here it makes sense to add average values for the seasons as well. Especially when you mention these seasons in the text, in Figures or in Table S2. And replace "media" by "median".

Figure 6. In what kind of software was the visualization of Figure 6 done? If in OpenAir, it should be quoted – see Carslaw and Ropkins (2012).

Figure 7. In what kind of software was the visualization of Figure 7 done? If in Zefir, it should be quoted - see Petit et al. (2017).

Table S1: This table provides only sketchy data, which are difficult to compare without context. First, it is necessary to distinguish between cities and countries. It is clear that, for example, Spain is smaller than China, but even so, there are different types of sites with different levels of concentrations (e.g., Querol et al., 2013; Sánchez de la Campa et al., 2009; Viana et al., 2006). The same for Italy... Second, similar sites and the same aerosol fraction should be compared (there is a difference in OC/EC ratio for PM2.5 and PM10). Different aerosol PM fractions should be mentioned in the table. There are usually also differences between seasons (typically winter vs. summer) so comparing different periods, for example, annual data with a month of winter data, is also little bit misleading. Third, if other temperature protocols are used in referenced studies, this should also be mentioned, as it also affects the OC/EC ratio. Contrary, if all the cited studies were analyzed by the TOT method, it is not necessary to repeat it in the table and it is enough just to mention it in a table legend. Figure S1: Indicate on which axis the online and offline data are displayed.

[Figure]

References:

Bautista VII, A.T., Pabroa, P.C.B., Santos, F.L., Quirit, L.L., Asis, J.L.B., Dy, M.A.K., Martinez, J.P.G., 2015. Intercomparison between NIOSH, IMPROVE_A, and EUSAAR_2 protocols: Finding an optimal thermal-optical protocol for Philippines OC/EC samples. Atmos. Pollut. Res. 6, 334–342. https://doi.org/10.5094/APR.2015.037

Carslaw, D.C., Ropkins, K., 2012. Openair - An r package for air quality data analysis. Environ. Model. Softw. 27–28, 52–61. https://doi.org/10.1016/j.envsoft.2011.09.008

Chen, L.-W.A., Chow, J.C., Wang, X.L., Robles, J.A., Sumlin, B.J., Lowenthal, D.H., Zimmermann, R., Watson, J.G., 2015. Multi-wavelength optical measurement to enhance thermal/optical analysis for carbonaceous aerosol. Atmos. Meas. Tech. 8, 451–461. https://doi.org/10.5194/amt-8-451-2015

Chow, J.C., Watson, J.G., Chen, L.-W.A., Chang, M.C.O., Robinson, N.F., Trimble, D., Kohl, S., 2007. The IMPROVE_A temperature protocol for thermal/optical carbon analysis: maintaining consistency with a long-term database. J. Air Waste Manage. Assoc. 57, 1014–1023. https://doi.org/10.3155/1047-3289.57.9.1014

Chow, J.C., Watson, J.G., Crow, D., Lowenthal, D.H., Merrifield, T., 2001. Comparison of IMPROVE and NIOSH Carbon Measurements. Aerosol Sci. Technol. 34, 23–34.

Chow, J.C., Watson, J.G., Green, M.C., Wang, X., Chen, L.W.A., Trimble, D.L., Cropper, P.M., Kohl, S.D., Gronstal, S.B., 2018. Separation of brown carbon from black carbon for IMPROVE and Chemical Speciation Network PM2.5 samples. J. Air Waste Manag. Assoc. 68, 494–510. https://doi.org/10.1080/10962247.2018.1426653

Jeong, C.-H., Hopke, P.K., Kim, E., Lee, D.-W., 2004. The comparison between thermal-optical transmittance elemental carbon and Aethalometer black carbon measured at multiple monitoring sites. Atmos. Environ. 38, 5193–5204. https://doi.org/10.1016/j.atmosenv.2004.02.065

Karanasiou, A., Panteliadis, P., Perez, N., Minguillón, M.C., Pandolfi, M., Titos, G.,

Viana, M., Moreno, T., Querol, X., Alastuey, A., 2020. Evaluation of the Semi-Continuous OCEC analyzer performance with the EUSAAR2 protocol. Sci. Total Environ. 747. https://doi.org/10.1016/j.scitotenv.2020.141266

Petit, J.E., Favez, O., Albinet, A., Canonaco, F., 2017. A user-friendly tool for comprehensive evaluation of the geographical origins of atmospheric pollution: Wind and trajectory analyses. Environ. Model. Softw. 88, 183–187. https://doi.org/10.1016/j.envsoft.2016.11.022

Petzold, A., Ogren, J.A., Fiebig, M., Laj, P., Li, S.M., Baltensperger, U., Holzer-Popp, T., Kinne, S., Pappalardo, G., Sugimoto, N., Wehrli, C., Wiedensohler, A., Zhang, X.Y., 2013. Recommendations for reporting black carbon measurements. Atmos. Chem. Phys. 13, 8365–8379. https://doi.org/10.5194/acp-13-8365-2013

Pio, C., Cerqueira, M., Harrison, R.M., Nunes, T., Mirante, F., Alves, C., Oliveira, C., Sanchez de la Campa, A., Artíñano, B., Matos, M., 2011. OC/EC ratio observations in Europe: Re-thinking the approach for apportionment between primary and secondary organic carbon. Atmos. Environ. 45, 6121–6132. https://doi.org/10.1016/j.atmosenv.2011.08.045

Querol, X., Alastuey, A., Viana, M., Moreno, T., Reche, C., Minguillón, M.C., Ripoll, A., Pandolfi, M., Amato, F., Karanasiou, A., Pérez, N., Pey, J., Cusack, M., Vásquez, R., Plana, F., DallOsto, M., de la Rosa, J., Sánchez de la Campa, A., Fernández-Camacho, R., Rodríguez, S., Pio, C., Alados-Arboledas, L., Titos, G., Artíñano, B., Salvador, P., García Dos Santos, S., Fernández Patier, R., 2013. Variability of carbonaceous aerosols in remote , rural , urban and industrial environments in Spain: implications for air quality policy. Atmos. Chem. Phys. 13, 6185–6206. https://doi.org/10.5194/acp-13-6185-2013

Sahu, M., Hu, S., Ryan, P.H., Le Masters, G., Grinshpun, S. a, Chow, J.C., Biswas, P., 2011. Chemical compositions and source identification of PM2.5 aerosols for estimation of a diesel source surrogate. Sci. Total Environ. 409, 2642–51.

https://doi.org/10.1016/j.scitotenv.2011.03.032

Sánchez de la Campa, A.M., Pio, C., de la Rosa, J.D., Querol, X., Alastuey, A., González-Castanedo, Y., 2009. Characterization and origin of EC and OC particulate matter near the Doñana National Park (SW Spain). Environ. Res. 109, 671–681. https://doi.org/10.1016/j.envres.2009.05.002

Sandradewi, J., Prevot, A.S.H., Szidat, S., Perron, N., Alfarra, M.R., Lanz, V.A., Weingartner, E., Baltensperger, U., 2008. Using Aerosol Light Absorption Measurements for the Quantitative Determination of Wood Burning and Traffic Emission Contributions to Particulate Matter. Environ. Sci. Technol. 42, 3316–3323.

Viana, M., Chi, X., Maenhaut, W., Querol, X., Alastuey, A., 2006. Organic and elemental carbon concentrations in carbonaceous aerosols during summer and winter sampling campaigns in Barcelona , Spain 40, 2180–2193. https://doi.org/10.1016/j.atmosenv.2005.12.001

Vodicka, P., Schwarz, J., Brus, D., Zdimal, V., 2020. Online measurements of very low elemental and organic carbon concentrations in aerosols at a subarctic remote station 226, 117380. https://doi.org/10.1016/j.atmosenv.2020.117380

Wu, C., Yu, J.Z., 2016. Determination of primary combustion source organic carbon-to-elemental carbon ( OC / EC ) ratio using ambient OC and EC measurements : secondary OC-EC correlation minimization method. Atmos. Chem. Phys. 16, 5453–5465. https://doi.org/10.5194/acp-16-5453-2016 Yan, C., Zheng, M., Shen, G., Cheng, Y., Ma, S., Sun, J., Cui, M., Zhang, F., Han, Y., Chen, Y., 2019. Characterization of carbon fractions in carbonaceous aerosols from typical fossil fuel combustion sources. Fuel 254, 115620. https://doi.org/10.1016/j.fuel.2019.115620

Zhu, C., Cao, J., Tsai, C., Shen, Z., 2014. Comparison and implications of PM2.5 carbon fractions in different environments. Sci. Total Environ. 466–467, 203–209. https://doi.org/10.1016/j.scitotenv.2013.07.029

Zikova, N., Vodicka, P., Ludwig, W., Hitzenberger, R., Schwarz, J., 2016. On the use of the field Sunset semi-continuous analyzer to measure equivalent black carbon concentrations. Aerosol Sci. Technol. 50, 284–296. https://doi.org/10.1080/02786826.2016.1146819
* * *

---

## Referee Comment (RC2) · Anonymous Referee #3 · 26 Jan 2021

As far as I understood it is for the first time real-time measurement of OC EC determination using a dual wavelength Sunset. The method is a milestone for a new technical to study EC or brown carbon. The paper is well structured and present a very new dataset which may be helpful for the scientific community. I recommend for a publication in AMT after they may address the following comments.

Method: because the most important work in this study should be new instrument set-up of new type of Sunset, I would suggest move this part to the very beginning part of method.

A typical thermogram of analysis including information of temperature, NDIR values

[Figure]

(CO2) and transmittance in two wavelengths should be added.

Figure 3: high dEC/OC was found in winter (Jan, Feb), whereas high OC/EC was found in late spring and summer. Such a different seasonal (and diurnal in Figure 4) trend indicate dEC/OC is not an indicator for SOC but rather an indicator of anthropogenic tracer. The seasonal variation of different carbonaceous should be discussed more carefully. Monthly and diurnal cycles of dEC may be added.

The source of dEC may be linked to BrC, but this remains unclear. I suggest the authors should include study outlook to resolve this problem.
* * *

---

## Author Comment (AC1) · 4 Mar 2021

Manuscript by Bao et al. describes the annual measurement of carbonaceous aerosols in Nanjing, China, using a two-wavelength Sunset Lab. semi-online analyzer. If I understand well, this is the first article where a semi-online Sunset EC/OC analyzer with a dual laser is presented. The manuscript thus shows both an improved instrument for measuring of EC/OC and a demonstration of the newly measured value dEC, which is presented as an alternative indicator to brown carbon (BrC) measurement. From this point of view, the publication in AMT is logical. However, I have a few comments and recommendations.

R: We thank the reviewer for the brief summary and positive comments on our paper.

General comments:

Since this is the first described use of a two-wavelength analyzer, the article should contain more information about instrument itself. It means how it differs from previous versions of the instrument, what software is used to evaluate data, and authors also should provide a Figure with an example of a typical analysis. What are the differences, for example, in comparison with the 7-wave offline DRI analyzer (Chen et al., 2015). It is also not clear from the description in the methodology how the split between OC and EC is determined. Are there any corrections used?

R: We appreciate the reviewer for the general suggestions on the description of our instrument. The most important change in our instrument was that "In this study, we modified the Sunset instrument to a two-wavelength (658 nm and 405 nm) Sunset carbon analyzer by adding one more violet diode laser at λ=405 nm. The violet diode laser together with the red diode laser, focus through the sample chamber then the laser beam passed through the filter to correct for the pyrolysis-induced error." This information was shown in lines 122-126 in the introduction section.

We agree with the reviewer that we need to add a figure with an example of a typical analysis. We added the sentence "The split point between OC and EC was detected by the RTCalc731 software provided by Sunset Lab. The principle was same as the traditional Sunset carbon analyzer (Birch and Cary, 1996). An example thermogram of sample analysis using the two-wavelength Sunset semi-continuous carbon analyzer was shown in Fig. 2. During the sample analysis, the laser beam at 658 nm and 405 nm were both sent through the filter and the transmitted light signal were monitored to correct the undesired formation of pyrolyzed carbon (PyrC) and then to determine the split point of OC and EC at both two wavelengths." in lines 170-176. The determination of the split point was conducted same as other researches, so we didn't do any corrections..

In our opinion, compared with the work reported by Chen et al. (2015), the most important meaning of our work is that we can provide the observation of long-term real-time dEC mass concentrations and the dEC data can be further applied in the future in the research of BrC. We added the sentence in lines 126-131 as "Previous work reported by Chen et al. (2015) as mentioned above was integrating the optical instrument like the aethalometer to the traditional OC/EC analyzer, in this way, they provided the light absorption contributions of BC and BrC. The enhanced carbon analyzer provided new insight into more accurate OC and EC measurements. Their work was conducted in offline mode, based on their work, our instrument can get the real-time OC and EC mass concentrations both at 658 nm and 405 nm."

[Figure]

Figure 2. Example thermogram of sample analysis using the two-wavelength Sunset semi-continuous carbon analyzer.

How much of the dEC roughly overlaps with the pyrolyzed carbon (PyrC) that was determined at the EC658nm split? This information can be very useful for previous studies where the PyrC level has been reported.

R: The PyrC at 658 nm contributed 4.4% to dEC. We added the sentence "In order to evaluate the impact of PyrC, we calculated the PyrC at 658 nm fraction of dEC and the average PyrC/dEC was 4.4%, indicating the little influence of PyrC." in lines 192-194 in the revised manuscript.

What was the distribution of dEC within the EC fractions? This information would be particularly valuable for studies in which OC and EC fractions are used for PMF analyzes in determining sources (e.g., Sahu et al., 2011; Yan et al., 2019; Zhu et al., 2014).

R: We added the dEC/EC in Table 1 in the revised manuscript and added the description in lines 263-267 as "The average dEC mass concentration was 0.8 µg m$^{-3}$ contributing 10.0% to OC, 22.3% to EC and 1.3% to PM$_{2.5}$ concentrations with max concentration of 8.1 µg m$^{-3}$ contributing 48.2% to OC, 97.8% to EC and 17.6% to total PM$_{2.5}$ concentrations. This information can be further applied in the PMF analysis to evaluate the sources of carbonaceous aerosols (Zhu et al., 2014;Sahu et al., 2011;Yan et al., 2019)."

Many studies compare the relationship between BC and EC (e.g., Jeong et al., 2004; Karanasiou et al., 2020) to determine the relationship between quantities that are determined by different measurement methods. While BC reflects the optical properties of the aerosol, EC rather tends to define the chemical composition. However, the relationship between them undoubtedly exists (Petzold et al., 2013). An analogous relationship should be between the optically determined BrC and the thermo-optically determined dEC, which is newly defined by the authors of this manuscript. In the introduction to the manuscript, the authors outline relationship between BrC a dEC, but further in the text, they continue with this statement only as not proved hypothesis that dEC is an alternative to BrC. However, this relationship should be demonstrated from parallel BC data measured optically at different wavelengths, e.g., from aethalometer (Sandradewi et al., 2008). If the authors do not have parallel measured data from the aethalometer, then optically measured BC data can be obtained directly from the EC/OC analyzer measurements (Chen et al., 2015; Vodicka et al., 2020; Ziková et al., 2016). The subsequent determination of BrC can then be performed similarly to Chowet al. (2018).

R: We appreciate the reviewer for the very useful suggestions. We were regret that we didn't have the parallel BC data during the study period. However, we do have both the Aethalometer data and Sunset data in winter, 2019. To investigate the relationship between the optically determined BrC and the dEC concentrations, we chose one month data in December, 2019 and added a new 2.3 section in lines 195-215 in the revised manuscript:

**"2.3 Test of the new dEC data**

To evaluate the new dEC data, parallel BC concentrations were measured with a seven-wavelength Aethalometer with dEC concentrations in December, 2019. Radiation attenuation of an aerosol deposition on a filter (ATN$_\lambda$) is determined by the Beer-Lambert law:

$$ATN_\lambda = ln\frac{I_{0,\lambda}}{I_\lambda} \qquad \text{Equation. (1)}$$

Where I$_{0,\lambda}$ and I$_\lambda$ were the measured wavelength-specific laser reflectance signals. ATN$_\lambda$ is used to calculate the attenuation coefficient with Eq. (2):

$$b_{ATN} = \frac{A}{V} \qquad \text{Equation. (2)}$$

Where A was the filter area and V is the sampled air volume. Then a simplified two-component model was used to calculate the contribution of light attenuation to both BC and BrC (Chow et al., 2018;Chen et al., 2015;Sandradewi et al., 2008;Hareley et al., 2008):

$$b_{ATN}(\lambda) = q_{BC} \times \lambda^{-AAE_{BC}} + q_{BrC} \times \lambda^{-AAE_{BrC}} \qquad \text{Equation. (3)}$$

Where q$_{BC}$ and q$_{BrC}$ were fitting coefficients, AAE was the absorption Ångström exponent which represented the wavelength-dependent characteristics of light absorption capability of aerosols. The AAE of BC was assumed to be 1. Fitting coefficients in Eq. (3) were obtained for potential

AAE$_{BrC}$ between 1 and 8 by least square linear regression and the AAE$_{BrC}$ led to the overall best fit in terms of $r^2$ is selected as the effective AAE$_{BrC}$. Using these fitting coefficients, the b$_{ATN}$ due to BC and BrC are calculated at each wavelength. Figure S2 showed that the fitted b$_{ATN}$ at 405 nm were within ±5 % of the measured values for b$_{ATN}$ > 0.01. Figure 3 showed the relationship between the b$_{ATN}$ due to BrC at 405 nm and the dEC. Good correlation between them were found with R square of 0.64, indicating that dEC was associated with BrC.

[Figure]

**Figure 3.** Relationship between the b$_{ATN}$ due to BrC at 405 nm and the dEC concentrations.

[Figure]

**Figure S2.** Meaured b$_{ATN}$ at 405 nm compared with b$_{ATN}$ fitted from Eq. (3) using a two-component model."

Specific comments:

lines 82-85: "This method has been wildly used in present studies applied in the thermal–optical transmittance (TOT) Sunset carbon analyzer based on NIOSH protocol or thermal–optical reflectance (TOR) Desert Research Institute (DRI) carbon analyzer based on IMPROVE_A protocol (Ji et al., 2016)" First, rather "widely" than "wildly". Second, the sentence is not completely correct because it is possible to use both NIOSH and IMPROVE or another temperature protocol on both devices from Sunset or DRI. Third, in a referenced paper by Ji et al. (2016), there is nothing about the IMPROVE_A protocol. About IMPROVE_A protocol is a paper by Chow et al. (2007).

R: We agree with the reviewer's suggestions. We changed the sentence to "This method has been wildly used in present studies applied in the NIOSH protocol or IMPROVE_A protocol." And we added the referenced paper by Chow et al. (2007). (see lines 82-83)

line 110: "Wang et al. (2011) used a two-wavelength Aethalometer...." The original work in which this type of BC distinction between wood burning and traffic emission was used is Sandradewi et al. (2008), which should be noted. Further, a reference to the work of Wang et al. (2011) is not in references.

R: We appreciate the reviewer for providing the reference to this work originally reported. We have added a brief introduce of the work reported by Sandradewi et al. (2008) in lines 87-92 as "(Sandradewi et al., 2008) pointed out that light absorption measurements at different wavelength by the aethalometer can be used to quantify the contributions of wood combustion and traffic emissions to aerosols since wood smoke contains organic compounds which enhance the light absorption in ultraviolet wavelength. But traffic emissions produce more BC, which dominates the light absorption in near-infrared wavelength. They took use of aethalometer data measured at 470 nm and 950 nm to quantify the BC distinction between wood burning and traffic emission."

We have added the reference by Wang et al. (2011) in references. (see lines 608-610)

line 102: "Italian Apennines" are quite broad area. Massabó specifically states the Ligurian Apennines, Italy. By the way, reference to the work of Massabó et al. (2016) disappeared from references in second version of manuscript.

R: We have changed the "the Italian Apennines" to "the Ligurian Apennines in Italy" and added the disappeared reference by Massabó et al. (2016). (see lines 106-107 and lines 565-567)

line 116: "important situ data" change to "important in-situ data"

R: Corrected. (see line 120)

line 148: "January-April in 2017" Why the authors report periods that they do not use in the evaluated data?

R: We apology for the wrong date we reported. The right periods were from August, 2015 to July, 2016 and we corrected the right periods in the revised manuscript. (see lines 219-220)

line 153: ". . .collected on prebaked quartz fiber filters. . ." Indicate temperature and time of filter prebaking.

R: Corrected. (see lines 225-226)

line 170: "of ~ 17mm" Filter diameter in Model-4 from Sunset Lab. is usually 16 mm. Is it different in the new type of device with two lasers?

R: We measured the diameter of filter used in our Sunset carbon analyzer and it is 17mm. According to the manual of the semi-continous OCEC carbon aerosol analyzer provided by Sunset Lab, the diameter of the filter is also 17mm. Below is the screenshot from the manual:

[Figure]

5. Using filter punch, cut a new 17mm quartz filter.

[Figure]

line 172: "modified NIOSH 5040 protocol" If the used NIOSH protocol was modified, the authors should describe it in detail (step, time, temperature) or give a reference where the protocol is described. In addition, the authors should explain why they chose NIOSH protocol which usually underestimate the EC (e.g., Bautista VII et al., 2015; Chow et al., 2001).

R: We thank the reviewer for asking about the protocol used in our instrument. The protocol we used is the default "RT-NIOSH 5040" protocol and we have added this table to the revised supplement as Table S1.

We thank the review for pointing out the disadvantage of NIOSH protocol and we will use different protocol in the future to provide more information about the results using our new two-wavelength instrument.

Table S1. Temperature protocol of the modified NIOSH 5040 method used in this study.

| Gas | Temperature(°C) | Time(s) |
|---|---|---|
| He-1 | 310 | 70 |
| He-2 | 480 | 60 |
| He-3 | 615 | 60 |
| He-4 | 840 | 100 |
| He/O$_2$-1 | 550 | 45 |
| He/O$_2$-2 | 625 | 45 |
| He/O$_2$-3 | 700 | 45 |
| He/O$_2$-4 | 775 | 45 |
| He/O$_2$-5 | 850 | 120 |
| CH$_4$/He | 0 | 120 |

line 193: "We also did the measurements of OC and EC in PM 2.5 filter samples using the same method followed by the NIOSH protocol." Were offline measurements also performed on a two-wavelength analyzer? If yes, please provide a comparison of the dEC. Second, were both data sets corrected (or uncorrected) to blank measurements before comparison?

R: First, we used the OCEC1028 software when doing the offline measurement. It was an old version software used in the traditional one-laser equipped Sunset carbon analyzer. We did the comparison between the online and offline data only to make sure the real-time OC and EC data were comparable, so we didn't consider the comparison between the dEC.  Second, we appreciate the reviewer for the reminding of the blank correction. All the data were corrected to blank measurement before comparison and we added the sentence "All the data were corrected to blank measurement before comparison." (see lines 188-189)

line 202: "(Draxler and Hess, 1998)" Authors of HYSPLIT prefer newer citations to their model - see: https://www.ready.noaa.gov/HYSPLIT_traj.php

R: We thank the reviewer for reminding the newer citations of HYSPLIT and we have added the new references (Cohen et al., 2015;Rolph et al., 2017). (see lines 241-242, lines 516-518 and lines line 233: "The average OC/EC ratios in this study was 3.6, which was lower than most of those reported in other studies..." This ratio depends, among other things, on the protocol used. If you are comparing with other studies, they need to use also the NIOSH protocol. If you compare with studies where they use, for example, the IMPROVE protocol, which generally analyzes higher concentrations of EC than the NIOSH protocol, then the OC/EC ratio between these studies is different.

R: We rechecked these studies reported by the other researchers, most of them used the NIOSH protocol. Still three of them including the work conducted in Hongkong, Italy and Mountain Tai used other protocols. Considering we displayed other similar area to these three sites, for example, Mountain Heng was similar with Mountain Tai which represented background sites and to compare with our study, we removed these studies and added the explain in the title of Table S2 as "Table S2. Comparisons of the concentrations of OC and EC in $PM_{2.5}$ between different cities in China and around the world using the TOT method applied in the NIOSH 5040 protocol."

lines 271-273: "The OC/EC ratio could give some information about primary and secondary organic carbon (Turpin and Huntzicker, 1995; Lim and Turpin, 2002)" The OC/EC ratio can be a rough indicator of the presence of primary and secondary carbon aerosols. Usually, this analysis is based on the determination of the OC/ECpri ratio and is applicable under certain conditions (Pio et al., 2011). If the authors want to discuss
the share of primary and secondary OCs, they should use some more recent approach– see, e.g., Wu and Yu (2016).

R: We agree with the reviewer that this is a rough discussion and added the sentence "It should be noted that the OC/EC ratios were a rough indicator to estimate the primary and secondary organic carbon, further analysis of the formation of SOC need to be conducted in the future (Pio et al., 2011; Wu and Yu, 2016)." at the end of the paragraph in lines 327-329. Considering this is the first time we report the work about the new two-wavelength method, this paper focused on the measurement and qualitatively discussion on the sources of dEC, so we did not do much work on the quantitatively discussion. We will conduct the work about the primary and secondary OCs in the future.

line 294: "...shown in Fig. 5. We also found similar distributions in dEC/OC and OC/EC." Not clear what authors mean by similar distribution in dEC/OC and OC/EC. In Fig. 5, color maps of these two parameters are different. Moreover, the conclusions that authors draw from it are not very clear from Figure 5. Maybe some other depiction of temperature and RH dependence would be more appropriate.

R: We agree with the reviewer. It is not appropriate to say that dEC/OC and OC/EC have the similar distribution. We removed the sentence "We also found similar distributions in dEC/OC and OC/EC." The high dEC/OC could be found in three area and the first area was similar with the distribution of OC/EC. We changed the sentence to "High dEC/OC (>30 %) could be found in three areas, first showed in the right area with relatively high T at 25-40 ℃ and RH at 40-60 %, which were usually found in the summer afternoon which was closely related to the strong formation of SOC. This distribution was also shown in OC/EC." (see lines 346-349)

We added two conclusions in the revised manuscript. First was shown in lines 352-353 as "In general, dEC had no strong dependence on the RH and T distribution, indicating the complex formation mechanism of dEC." Second was shown in lines 353-358 as "The OC and EC showed similar distributions with the highest mass loading (OC: > 20 µg m$^{-3}$; EC: > 8 µg m$^{-3}$) at relatively high RH at 60-80 % which usually occurred at night with relatively low boundary layer height, leading to the accumulation of aerosols. However the corresponding OC/EC ratios were low, suggesting the importance of primary sources to OC and EC in northern Nanjing, which will be verified in the wind rose of OC and EC (Fig. 8)."

line 303: Replace "aass" by "mass".
R: Corrected. (see line 359)

line 323: Replace "Local" by "local".
R: Corrected. (see line 379)

Table 1: Here it makes sense to add average values for the seasons as well. Especially when you mention these seasons in the text, in Figures or in Table S2. And replace "media" by "median".
R: Corrected. We agree with the reviewer and we added the seasonal mean values to Table 1 in the revised manuscript.

Figure 6. In what kind of software was the visualization of Figure 6 done? If in OpenAir, it should be quoted – see Carslaw and Ropkins (2012).

R: Yes, we have added the reference. (see line 362 and lines 493-494)

Figure 7. In what kind of software was the visualization of Figure 7 done? If in Zefir, it should be quoted - see Petit et al. (2017).

R: Yes, we have added the reference. (see line 374 and lines 572-574)

Table S1: This table provides only sketchy data, which are difficult to compare without context. First, it is necessary to distinguish between cities and countries. It is clear that, for example, Spain is smaller than China, but even so, there are different types of sites with different levels of concentrations (e.g., Querol et al., 2013; Sánchez de la Campa et al., 2009; Viana et al., 2006). The same for Italy. . . Second, similar sites and the same aerosol fraction should be compared (there is a difference in OC/EC ratio for PM2.5 and PM10). Different aerosol PM fractions should be mentioned in the table. There are usually also differences between seasons (typically winter vs. summer) so comparing different periods, for example, annual data with a month of winter data, is also little bit misleading. Third, if other temperature protocols are used in referenced studies, this should also be mentioned, as it also affects the OC/EC ratio. Contrary, if all the cited studies were analyzed by the TOT method, it is not necessary to repeat it in the table and it is enough just to mention it in a table legend.

R: We Thank the reviewer for the suggestions. We distinguished the cities and countries and added the site type description. Since all the references displayed in the table collected $PM_{2.5}$ and used the NIOSH 5040 protocol, we changed the title to "Comparisons of the concentrations of OC and EC in $PM_{2.5}$ between different cities in China and around the world using the TOT method applied in the NIOSH 5040 protocol."

[revised manuscript text omitted]

China and around the world using the TOT method applied in the NIOSH 5040 protocol.

| Country | City or region | Site type | Sampling period | OC | EC | OC/ EC | References |
|---|---|---|---|---|---|---|---|
| China | Beijing | Urban | Mar 2013-Feb 2014 | 14.0 | 4.1 | 3.4 | (Ji et al., 2016) |
| China | Shanghai | Urban | Oct 2005-Jul 2006 | 14.7 | 2.8 | 5.0 | (Feng et al., 2009) |
| China | Chengdu | Urban | May 2012-Apr 2013 | 19.0 | 4.6 | 4.3 | (Chen et al., 2014) |
| China | Chongqing | Urban | May 2012-Apr 2013 | 15.2 | 4.0 | 3.8 | (Chen et al., 2014) |
| China | Nanjing | Suburban | Annual 2014 | 5.7 | 3.2 | 1.8 | (Chen et al., 2017) |
| China | Guangzhou | Rural | Mar 2012–Feb 2013 | 6.1 | 0.8 | | (Lai et al., 2016) |
| China | Mount Heng | Set at 1269 m asl | Mar-May 2009 | 3.0 | 0.5 | 5.2 | (Zhou et al., 2012) |
| Mexico | Mexico City | Suburban | Mar 2006 | 6.4 | 2.1 | 4.5 | (Yu et al., 2009) |
| India | Delhi | Suburban | Nov 2010-Feb 2011 | 54.1 | 10.4 | 5.2 | (Tiwari et al., 2012) |
| America | Philadelphia | Suburban | Jul 2002-Aug 2002 | 4.8 | 0.4 | 18.7 | (Jeong et al., 2004) |
| America | Rochester | Suburban | Jun 2002 | 9.2 | 0.3 | 23.6 | (Jeong et al., 2004) |
| Spain | Aragón | Urban | Dec 2011 | 3.6 | 1.1 | 4.7 | (Escudero et al., 2015) |
| China | Nanjing | Suburban | Jun 2015-Aug 2016 | 8.6 | 2.9 | 3.6 | This study |

**Table S3.** Statistics on the meteorological factors in four seasons at NUIST site during the study
period.

| | Atmospheric Pressure (hPa) | Relative Humidity (%) | Temperature (°C) | Wind Speed (m s$^{-1}$) | Total Precipitation (mm) |
|---|---|---|---|---|---|
| Spring | 1009.9 | 66.0 | 16.8 | 1.9 | 256.3 |
| Summer | 1000.7 | 72.6 | 26.7 | 1.4 | 586.0 |
| Autumn | 1014.6 | 71.0 | 19.5 | 1.7 | 218.5 |
| Winter | 1027.0 | 63.9 | 5.7 | 1.7 | 82.1 |

[Figure]

**Figure S1.** Correlations between the real-time OC, EC and TC concentrations (y-axis) and sampling OC, EC and TC concentrations (x-axis) during the corresponding periods.

[Figure]

**Figure S2.** Meaured $b_{ATN}$ at 405 nm compared with $b_{ATN}$ fitted from Eq. (3) using a two-component model.

[Figure]

**Figure S3.** dEC/OC variation at different intervals of OC/EC ratios in spring (a), summer (b),
autumn (c) and winter (d).

[Figure]

**Figure S4.** Time variations of OC, EC, dEC, dEC/OC, OC/EC and fire points obtained from the
Fire Information for Resource Management System (FIRMS) derived from the Moderate
Resolution Imaging Spectroradiometer (MODIS).

[Figure]

**Figure S5.** 48-h back trajectories at 500 m from the study site from 8 June 2015 to 9 June 2015(a),

June 2015 to 12 June 2015 (b), respectively and from 7 February 2016 to 10 February 2016 (c)

and 26 February 2016 to 27 February 2016 (d), respectively.

---

## Author Comment (AC2) · 4 Mar 2021

As far as I understood it is for the first time real-time measurement of OC EC determination using a dual wavelength Sunset. The method is a milestone for a new technical to study EC or brown carbon. The paper is well structured and present a very new dataset which may be helpful for the scientific community. I recommend for a publication in AMT after they may address the following comments.

R: We thank the reviewer for the brief summary and positive comments on our paper.

Method: because the most important work in this study should be new instrument setup of new type of Sunset, I would suggest move this part to the very beginning part of method.

R: We thank the reviewer for the suggestion. Considering the site description was also very important where all the real-time observations and sampling were conducted, we kept the site description in the front and put the "Two-wavelength TOT measurement" after the site description.

A typical thermogram of analysis including information of temperature, NDIR values (CO2) and transmittance in two wavelengths should be added.

R: We agree with the reviewer that we need to add a figure with an example of a typical analysis. We added the sentence "The split point between OC and EC was detected by the RTCalc731 software provided by Sunset Lab. The principle was same as the traditional Sunset carbon analyzer (Birch and Cary, 1996). An example thermogram of sample analysis using the two-wavelength Sunset semi-continuous carbon analyzer was shown in Fig. 2. During the sample analysis, the laser beam at 658 nm and 405 nm were both sent through the filter and the transmitted light signal were monitored to correct the undesired formation of pyrolyzed carbon (PyrC) and then to determine the split point of OC and EC at both two wavelengths." in lines 170-176.

**Figure 2.** Example thermogram of sample analysis using the two-wavelength sunset semicontinuous carbon analyzer.

Figure 3: high dEC/OC was found in winter (Jan, Feb), whereas high OC/EC was found in late spring and summer. Such a different seasonal (and diurnal in Figure 4) trend indicate dEC/OC is not an indicator for SOC but rather an indicator of anthropogenic tracer. The seasonal variation of different carbonaceous should be discussed more carefully. Monthly and diurnal cycles of dEC may be added.

R: We thank the reviewer for the suggestions. We added the monthly and diurnal cycles of dEC in Figure 5 and Figure 6 in the revised manuscript and the sentence "High dEC/OC was found in January and February in winter, indicating strong influence of anthropogenic sources on dEC, such as coal combustion. In addition, we found strong biomass burning activities in February, which

significantly contributed to the high concentrations of dEC in February, more details could be found in section 3.3." in lines 306-309 in the revised manuscript. We added the sentence "Similar though not so obvious diurnal variations were found in dEC." in lines 339-340.

Our results did show that the sources of dEC were complicated. Anthropogenic sources could contribute to dEC. In conclusion, we pointed out that "The results showed that high (low) OC, EC and dEC concentrations were found in Winter (summer), indicating the significant impact of the increase of various emission sources in winter and wet scavenging of rain in summer." and "It should be noted that the sources of dEC were complicated and the anthropogenic emissions and secondary formations of dEC aerosols couldn't be ignored , further chemical analysis need to be conducted in the future. We also hope that the dEC data can be further applied in more researches." Using the MODIS fire information and receptor model, we proved that biomass burning significantly contributed to high dEC concentrations. However, with the limited data, the anthropogenic or secondary sources of dEC couldn't be quantified. We hope further analysis of this work can be done in the future.

The source of dEC may be linked to BrC, but this remains unclear. I suggest the authors should include study outlook to resolve this problem.

R: We added the sentence "The evaluation of SOC formation and the relationship between dEC and SOC can be conducted. In addition, More chemical analysis such as the analysis of the ion, the organic matter or the sugars in  $PM_{2.5}$  can be measured, thus we can get some information of the tracers of different sources and more accurate and quantitative source apportionment can be done (Bhattaraia et al., 2019;Wu et al., 2019;Wu et al., 2018). We also hope that the dEC data can be further applied in more researches." at the end of the revised manuscript. (see lines 459-464)

Highly time-resolved characterization of carbonaceous aerosols using a two-wavelength
 Sunset thermo/optical carbon analyzer

3

Mengying Bao1,2,3, Yan-Lin Zhang1,2,3\*, Fang Cao1,2,3, Yu-Chi Lin1,2,3, Yuhang Wang4, Xiaoyan
Liu1,2,3, Wenqi Zhang1,2,3, Meiyi Fan1,2,3, Feng Xie1,2,3, Robert Cary5, Joshua Dixon5 and Lihua
Zhou6

[revised manuscript text omitted]

---

## Author Response (AR2)

We thank the editor for the comments which have helped us to improve the manuscript. Our point-by-point responses are below. The editor's comments are in black font and our responses are in blue font.

Comments to the Author:

The authors have reasonably addressed the comments of the two anonymous referees and they have modified their manuscript accordingly. However, the current version of the manuscript suffers from poor language and grammar and contains many errors and other shortcomings. Therefore, I have numerous alterations and corrections that should be made before the manuscript can be published in AMT.

For the Main text:

Use is made of both "thermo/optical" and "thermo-optical" within the manuscript. I strongly suggest that the authors stick to a single term and that, in addition, they use "thermal-optical" throughout the text (NOT in the References).

R: We have used the "thermal-optical" throughout the revised manuscript. (see lines 2, 29, 50-51, 76, 85, 101-102, 114, 117, and 453)

Line 24: remove the comma before " mainly".

R: Corrected. (see line 24)

Line 26: replace ", makes" by " and makes".

R: Corrected. (see line 26)

Line 30: replace "on one-year" by "on a one-year".

R: Corrected. (see line 30)

Line 31: replace "Due to" by "Since".

R: Corrected. (see line 31)

Line 34: replace "carbonaceous aerosols" by "carbonaceous aerosol".

R: Corrected. (see line 35)

Line 37: replace "in OC and EC aerosols, however dEC aerosols were" by "for OC and EC; however dEC was".

R: Corrected. (see line 37)

Line 41: replace "obvious higher" by "obviously higher".

R: Corrected. (see line 42)

Line 42: replace "days," by "days;".

R: Corrected. (see line 42)

Line 46: replace "monitored in winter," by "observed in winter and".

R: Corrected. (see line 46)

Line 46: abbreviations an…d acronyms, here "YRD", should be defined (written full-out) when first used. Since "YRD" is only used once within the Abstract, it should not be defined here, but replaced by "Yangtze River Delta".

R: Corrected. (see line 47)

Line 50: replace "traditional" by "the traditional".

R: Corrected. (see line 50)

Line 51: replace "BrC, the application of dEC data need" by "BrC; the application of dEC data needs".

R: Corrected. (see lines 51-52)

Line 69: replace "can't" by "cannot".

R: Corrected. (see line 70)

Line 70: replace "emitted from" by "produced by".

R: Corrected. (see line 71)

Line 73: replace "of BrC" by "of the BrC".

R: Corrected. (see line 74)

Line 74: replace "modeling method" by "modeling".

R: Corrected. (see line 75)

Line 76: replace "method of OC" by "methods for OC".

R: Corrected. (see line 77)

Line 79: replace "and return" by "and the return".

R: Corrected. (see line 80)

Line 80: replace "as a split" by "as the split".

R: Corrected. (see line 81)

Line 82: replace "wildly used in present studies applied in" by "widely used in studies employing".

R: Corrected. (see line 83)

Line 90: replace "in ultraviolet" by "in the ultraviolet".

R: Corrected. (see line 92)

Line 90: abbreviations and acronyms, here "BC", should be defined (written full-out) when first used within the main text.

R: Corrected. (see line 92)

Line 91: replace "in near-infrared" by "in the near-infrared".

R: Corrected. (see line 93)

Line 98: replace "(2012a) and Wang et al. (2012b)" by "(2012a; 2012b)".

R: Corrected. (see line 100)

Line 99: abbreviations and acronyms, here "TOT" and "TOR", should be defined (written full-out) when first used within the main text.

R: Corrected. (see lines 101-102)

Line 106: replace "collected wintertime" by "collected during wintertime".

R: Corrected. (see line109)

Line 114: replace "organic carbon (OC)" by "OC"; "OC" was already defined in line 53.

R: Corrected. (see line 116)

Line 115: replace "can't" by "cannot".

R: Corrected. (see line 117)

Line 118: replace "where cover" by "with".

R: Corrected. (see line 120)

Line 120: replace "EC aerosols" by "EC".

R: Corrected. (see line 123)

Line 124: replace "laser, focus" by "laser focus".

R: Corrected. (see line 127)

Line 125: replace "chamber then the laser beam passed" by "chamber, then the laser beam passes".

R: Corrected. (see line 127)

Line 127: replace "analyzer," by "analyzer;".

R: Corrected. (see line 130)

Line 129: replace "mode," by "mode;".

R: Corrected. (see line 132)

Line 137: replace "dEC aerosols" by "dEC".

R: Corrected. (see line 139)

Line 138: replace "observed" by "measured".

R: Corrected. (see line 139)

Line 141: replace "dEC aerosols" by "dEC".

R: Corrected. (see line 143)

Line 159: replace "analyzed" by "determined".

R: Corrected. (see line 160)

Line 160: replace "and applied" by "by applying".

R: Corrected. (see line 160)

Line 161: replace "were shown" by "are shown".

R: Corrected. (see line 161)

Line 162: replace "was first" by "is first".

R: Corrected. (see line 163)

Line 163: replace "was volatilized" by "is volatilized".

R: Corrected. (see line 164)

Line 164: replace "kept at 840°C for a while and went down to 550°C. In the second stage, EC was" by "is kept at 840°C for a while and goes down to 550°C. In the second stage, EC is".

R: Corrected. (see lines 164-165)

Line 166: replace "2%oxygen and 98%helium. The pyrolysis products were" by "2% oxygen and 98% helium. The pyrolysis products are".

R: Corrected. (see lines 166-167)

Line 168: replace "two-diode lasers (658nm and 405nm) equipped Sunset analyzer," by "a two-diode lasers (658 nm and 405 nm) equipped Sunset analyzer;".

R: Corrected. (see line 169)

Line 170: replace "was detected" by "is detected".

R: Corrected. (see line 171)

Line 171: replace "was same as the" by "is the same as for the".

R: Corrected. (see line 172)

Line 173: replace "was shown" by "is shown".

R: Corrected. (see line 174)

Line 174: replace "were both" by "are both" and replace "signal were" by "signal is".

R: Corrected. (see line 175)

Line 176: replace "both two" by "both".

R: Corrected. (see line 177)

Line 177: replace "658nm" by "658 nm" and replace "BrC in" by "BrC at the".

R: Corrected. (see line 178)

Line 178: replace "thus this enhanced absorption at 405nm" by "the enhanced absorption at 405 nm".

R: Corrected. (see line 179)

Line 180: replace "study provided" by "study provides".

R: Corrected. (see line 181)

Line 182: replace "658nm" by "658 nm".

R: Corrected. (see line 183)

Line 183: replace "5%methane" by "5% methane".

R: Corrected. (see line 184)

Line 184: replace "95%Helium was injected and thus a known carbon mass could" by "95% helium is injected and thus a known carbon mass can".

R: Corrected. (see line 185)

Line 188: replace "All the data" by "All data".

R: Corrected. (see line 189)

Line 192: insert a space before "of 0.4 for EC".

R: Corrected. (see line 193)

Lines 199, 202, and 206: remove "Equation.".

R: Corrected. (see lines 200, 203, and 207)

Line 200: replace "were the" by "are the".

R: Corrected. (see line 201)

Line 203: replace "was the" by "is the".

R: Corrected. (see line 204)

Line 204: replace "was used" by "is used".

R: Corrected. (see line 205)

Lines 205, 241, 267, 329, and 463: there should be a space after each ";".

R: Corrected. (see lines 205-206, 241, 266, 327, and 460)

Line 207: replace "were fitting" by "are fitting" and replace "was the" by "is the".

R: Corrected. (see line 208)

Line 208: replace "represented" by "represents".

R: Corrected. (see line 209)

Line 210: replace "square" by "squares" and replace "led to" by "leading to".

R: Corrected. (see line 211)

Line 212: replace "showed that" by "shows that".

R: Corrected. (see line 213)

Line 213: replace "were within" by "are within" and replace "showed the" by "shows the".

R: Corrected. (see lines 213-214)

Line 214: replace "were found" by "is found".

R: Corrected. (see line 215)

Lines 225-226: replace "were collected on prebaked quartz fiber filters which were under 450℃ for 6 hours (QFF, PALL, America) with 8*10 inch by" by "was collected on 8*10 inch prebaked quartz fiber filters (QFF, PALL, America) by".

R: Corrected. (see lines 226-227)

Lines 230-231: replace "bank filters in four seasons collected following 10 mins exposures" by "blanks in the four seasons using 10 min exposure".

R: Corrected. (see line 231)

Line 233: replace "weighed by" by "weighed with an".

R: Corrected. (see line 233)

Line 235: replace "for further" by "until further".

R: Corrected. (see line 235)

Line 239: replace "(NOAA) were" by "(NOAA), was".

R: Corrected. (see line 239)

Line 242: replace "air masses" by "air mass".

R: Corrected. (see line 242)

Line 243: replace "500m" by "500 m".

R: Corrected. (see line 243)

Lines 249-250: replace "be seen in the research reported by Bao" by "be found in Bao".

R: Corrected. (see line 249)

Line 257: replace "were comparable" by "are comparable".

R: Corrected. (see line 256)

Line 257: "Chen et al. (2017)" is missing in the Reference list.

R: We have added "Chen et al. (2017)" in the Reference list. (see lines 495-497)

Line 260: replace "were probably" by "are probably".

R: Corrected. (see line 259)

Line 264: replace "1.3% to" by "1.3% to the" and replace "with max" by "with maximum".

R: Corrected. (see line 263)

Line 266: replace "sources of" by "sources of the".

R: Corrected. (see line 265)

Line 268: replace "aerosols levels" by "aerosol levels".

R: Corrected. (see line 267)

Line 270: replace "was affected" by "are affected".

R: Corrected. (see line 269)

Line 272: replace "which was" by "which is".

R: Corrected. (see line 271)

Line 273: replace "set in" by "in".

R: Corrected. (see line 272)

Line 274: replace "aerosols concentrations in China was" by "aerosol concentrations in China is".

R: Corrected. (see line 273)

Line 276: replace "winter, the" by "winter and the".

R: Corrected. (see line 275)

Line 277: replace "ratios in this study was 3.6, which was" by "ratio in this study was 3.6, which is".

R: Corrected. (see line 276)

Line 278: replace "in our" by "at our".

R: Corrected. (see line 277)

Line 281: replace "periods. During the study" by "period. During that".

R: Corrected. (see line 279)

Line 285: replace "Larger" by "A larger".

Line 286: replace "for period" by "for".

Line 288: replace "carbonaceous aerosols mass fraction of 5.2%. The result indicated" by "a carbonaceous aerosol mass fraction of 5.2%. The results indicate that".

Line 289: replace "contributes" by "contribute".

Line 293: replace "14.3% when" by "14.3% for".

Line 294: replace "Similar trend was found in" by "A similar trend was found for the".

Line 295: replace "studies had" by "studies have".

Line 298: replace "in four" by "in the four".

Line 309: replace "February, more details could" by "February; more details can".

Line 310: replace "emissions intensities" by "emission intensities".

Line 311: replace "in four" by "in the four".

Line 312: replace "aerosols concentrations" by "aerosol concentrations".

Line 315: replace "summer, higher" by "summer, the higher".

Line 316: replace "and higher" by "and the higher".

Line 317: replace "into gaseous" by "into the gaseous".

Line 320: replace "The OC/EC" by "The average OC/EC".

R: Corrected. (see line 318)

Line 326: replace "the 3.3 sections" by "section 3.3".

R: Corrected. (see line 324)

Lines 326-327: replace "indicated that strong" by "indicate strong".

R: Corrected. (see lines 324-325)

Line 328: replace "carbon, further" by "carbon; further".

R: Corrected. (see line 326)

Line 329: replace "need to" by "needs to".

R: Corrected. (see line 327)

Line 334: replace "EC aerosols" by "EC".

R: Corrected. (see line 332)

Line 335: replace "quality in the" by "quality in".

R: Corrected. (see line 333)

Line 337: replace "source on" by "sources on".

R: Corrected. (see line 335)

Line 341: replace "there was no" by "there were no".

R: Corrected. (see line 339)

Line 343: replace "aerosols formations" by "aerosol formation".

R: Corrected. (see line 341)

Line 347: replace "could be found in three areas, first showed" by "can be found in three areas, first shown".

R: Corrected. (see line 344)

Line 349: replace "also shown" by "also seen".

R: Corrected. (see line 346)

Line 350: replace "was displayed" by "is".

R: Corrected. (see line 347)

Line 351: replace "appeared when" by "appears for".

R: Corrected. (see lines 347-348)

Line 353: replace "The OC and EC showed" by "OC and EC show".

R: Corrected.  (see line 350)

Line 356: replace "However the" by "However, the".

R: Corrected. (see lines 352-353)

Line 361: replace "influences of air masses" by "influence of air mass".

R: Corrected. (see line 357)

Line 362: replace "in four seasons is illustrated" by "in the four seasons is shown".

R: Corrected. (see line 358)

Line 364: replace "indicating by" by "indicated by".

R: Corrected. (see line 360)

Line 364: abbreviations and acronyms, here "WS", should be defined (written full-out) when first used within the main text.

R: Corrected. (see line 360)

Line 367: replace "that sources" by "that the sources".

R: Corrected. (see line 363)

Line 369: replace "aerosols showed" by "showed".

R: Corrected. (see line 365)

Lines 369-370: replace "with the increasing of WS and highest dEC/OC were found when" by "with increasing WS and the highest dEC/OC were found for".

R: Corrected. (see line 366)

Line 371: replace "was highly likely the main sources contributing to" by "were highly likely the main sources contributing to the".

R: Corrected. (see line 367)

Line 373: replace "map are" by "maps are".

R: Corrected. (see line 370)

Line 375: replace "PSCF" by "the PSCF".

R: Corrected. (see line 372)

Line 376: replace "dEC aerosols and local contributions to OC and EC aerosols" by "dEC and local contributions to OC and EC".

R: Corrected. (see line 373)

Line 377: replace "Nanjing," by "Nanjing;".

R: Corrected. (see line 374)

Line 378: replace "dEC aerosols" by "dEC".

R: Corrected. (see line 375)

Line 379: replace "main sources" by "main source".

R: Corrected. (see line 376)

Line 380: replace "from southeast" by "from the southeast".

R: Corrected. (see line 377)

Line 383: replace "sources areas of dEC and EC were displayed" by "sources areas of dEC and EC were".

R: Corrected. (see lines 379-380)

Line 384: replace "to biomass burning," by "with biomass burning;".

R: Corrected. (see line 381)

Line 385-366: replace "were in the section 3.3. In autumn, strongest local sources from the study site of OC and EC were found. However, the dEC" by "are given in section 3.3. In autumn, local sources from the study site were strongest for OC and EC. However, dEC".

R: Corrected. (see lines 381-383)

Line 392: replace "aerosols pollution levels. In winter, dEC were" by "aerosol pollution levels. In winter, dEC was".

R: Corrected. (see lines 388-389)

Line 397: replace "scale, it" by "scale; it".

R: Corrected. (see line 394)

Line 399: replace "reached to" by "amounted to".

R: Corrected. (see line 396)

Lines 400-401: replace "Mar 2016 in the areas around our study site, respectively, suggesting" by "March 2016, respectively, in the areas around our study site, suggesting".

R: Corrected. (see lines 397-398)

Line 402: replace "dEC aerosols" by "dEC".

R: Corrected. (see line 399)

Line 403: replace "3 Mar" by "3 March".

R: Corrected. (see line 400)

Line 404: replace "aerosols concentrations" by "aerosol concentrations".

R: Corrected. (see line 401)

Line 407: replace "we did found" by "we found".

R: Corrected. (see line 404)

Line 409: replace "went through" by "passed over".

R: Corrected. (see line 406)

Line 410: replace "in northwest" by "in the northwest".

R: Corrected. (see line 407)

Line 411: replace "trajectory showed air mass" by "trajectories showed that the air masses".

R: Corrected. (see line 408)

Line 413: replace "trajectory showed" by "trajectories showed".

R: Corrected. (see line 411)

Line 414: replace "mass was" by "masses came".

R: Corrected. (see line 411)

Line 416: replace "periods which was" by "period which were".

R: Corrected. (see line 413)

Line 418: replace "appeared on" by "on".

R: Corrected. (see line 415)

Line 420: replace "only sources" by "only source" and replace "in the 3.1 and 3.2 section" by "in sections 3.1 and 3.2".

R: Corrected. (see line 417)

Line 421: replace "couldn't be ignored, too" by "cannot be ignored, either".

R: Corrected. (see line 418)

Line 424: replace "obvious higher" by "obviously higher" and replace "than those in" by "than in".

R: Corrected. (see lines 420-421)

Line 426: replace "dEC aerosols" by "dEC".

R: Corrected. (see line 423)

Line 428: replace "winter, in addition, the location" by "winter; in addition, the locations".

R: Corrected. (see lines 424-425)

Line 430: replace "in the section" by "in section".

R: Corrected. (see line 427)

Line 432: replace "Conclusion" by "Conclusions".

R: Corrected. (see line 429)

Line 436: replace "wavelength at" by "wavelength" and replace "so we" by "so that we".

R: Corrected. (see line 433)

Line 439: replace "Winter" by "winter".

R: Corrected. (see line 436)

Line 440: replace "of rain" by "by rain".

R: Corrected. (see line 437)

Line 442: replace "layers" by "layer".

R: Corrected. (see line 439)

Line 443: replace "significant" by "a significant".

R: Corrected. (see line 440)

Line 444: replace "ratios increased" by "ratio increased".

R: Corrected. (see line 441)

Lines 446-447: replace "in OC and EC aerosols, however dEC aerosols were" by "for OC and EC; however, dEC was".

R: Corrected. (see lines 443-444)

Line 450: replace "informations" by "information".

R: Corrected. (see line 446)

Line 451: replace "dEC aerosols in" by "dEC in".

R: Corrected. (see line 448)

Line 453: replace "Large" by "A large".

R: Corrected. (see line 449)

Lines 453-454: replace "were monitored, significantly contributed to all the carbonaceous aerosols pollutions" by "was observed; these fires significantly contributed to the carbonaceous aerosol pollution".

R: Corrected. (see lines 450-451)

Line 455: replace "than traditional" by "than the traditional".

R: Corrected. (see line 452)

Line 458: replace "couldn't be ignored , further" by "could not be ignored; further".

R: Corrected. (see lines 455-456)

Line 459: replace "need to" by "needs to".

R: Corrected. (see line 456)

Line 460: replace "More chemical" by "more chemical".

R: Corrected. (see lines 457-458)

Line 461: replace "of the ion, the organic matter or the sugars in PM2.5 can be measured," by "for ions, organic matter or sugars in PM2.5 can be made;".

R: Corrected. (see line 458)

Line 463: replace "Wu et al., 2018;Wu et al., 2019" by "Wu et al., 2018; 2019".

R: Corrected. (see line 460)

Line 464: replace "researches" by "research".

R: Corrected. (see line 461)

Line 467: replace "is financially" by "was financially".

R: Corrected. (see line 464)

Line 470: replace "is supported" by "was supported".

R: Corrected. (see line 467)

Line 472: insert a space before "We would".

R: Corrected. (see line 469)

Line 474: replace "obsevation period. Besides, we are grateful for Prof. Yunhua Chang, who makes" by "observation period. Besides, we are grateful for Prof. Yunhua Chang, who made".

R: Corrected. (see line 471)

Line 475: replace "suggerestions to" by "suggestions for".

R: Corrected. (see line 472)

Line 636: replace "7," by "7, 43182,".

R: Corrected. (see line 636)

Line 639: replace "254," by "254, 115620,".

R: Corrected. (see line 639)

Line 655: replace "38," by "38, L21810,".

R: Corrected. (see line 655)

Line 677, inside the figure: replace twice "solit" by "split".

R: Corrected. (see line 677)

Line 700: replace "periods" by "period".

R: Corrected. (see line 700)

Line 709: replace twice "bar" by "bars".

R: Corrected. (see line 709)

Line 712: replace "3 Mar" by "3 March".

R: Corrected. (see line 712)

Line 713: replace twice "bar" by "bars".

R: Corrected. (see line 713)

Line 719, inside the table: replace "Mar" by "March".

R: Corrected. (see line 719)

For the Supplement:

Page 1: At the top of this page it should be indicated that this the Supplement; furthermore, the title of the paper and at least its first author should be given.

R: We have added a title of the Supplement including the title of the paper and the first author. (see lines 1-4)

Line 5: replace "Comparisons" by "Comparison".

R: Corrected. (see line 10)

Line 7: replace "at NUIST" by "at the NUIST".

R: Corrected. (see line 12)

Line 13: replace "Meaured" by "Measured".

R: Corrected. (see line 18)

Line 14: replace "model" by "model.".

R: Corrected. (see line 19)

Line 20: replace "from the" by "above the".

R: Corrected. (see line 25)

Line 24: replace "Reference" by "References".

R: Corrected. (see line 29)

Line 28: replace "Comparisons" by "Comparison".

R: Corrected. (see line 33)

Line 31: replace "at NUIST" by "at the NUIST".

R: Corrected. (see line 36)

Line 39: replace "Meaured" by "Measured".

R: Corrected. (see line 44)

Line 50: replace "from the" by "above the".

R: Corrected. (see line 55)

Line 69: replace "Ji, D., Zhang, J., He, J., Wang, X., BoPanga, Liua, Z., Wang, L., and Wang, Y.:" by "Ji, D. S., Zhang, J. K., He, J., Wang, X. J., Pang, B., Liu, Z. R., Wang, L. L., and Wang, Y. S.:"

R: Corrected. (see line 74)

Line 82: an abbreviated journal name is needed here.

R: Corrected. (see line 87)

[revised manuscript text omitted]

---

## Author Response (AR3)

**Response to editor's comments**

We thank the editor for the comments which have helped us to improve the manuscript. Our point by-point responses are below. The editor's comments are in black font and our responses are in blue font.

Comments to the Author:

The following alterations are still needed in the Main text before the manuscript can be published in AMT:

Line 47: replace "Yangtze River Delta (YRD) area" by "Yangtze River Delta area".

R: corrected.

Line 77: replace "thermo-optical" by "thermal-optical".

R: corrected.

Line 83: replace "wildly used" by "widely used".

R: corrected.

Line 184: replace "95% Helium" by "95% helium".

R: corrected.

Line 230: replace "banks in" by "blanks in".

R: corrected.

Line 323: replace "in 3.3 sections" by "in section 3.3".

R: corrected.

Line 378: replace "sources areas" by "source areas".

R: corrected.

Line 509: replace "Air Waste." by "Air Waste".

R: corrected.